# TIGERSCORE: TOWARDS BUILDING EXPLAINABLE METRIC FOR ALL TEXT GENERATION TASKS

## ABSTRACT

We present TIGERScore, a **T**rained metric that follows **I**nstruction **G**uidance to perform **E**xplainable, and **R**eference-free evaluation over a wide spectrum of text generation tasks. Different from other automatic evaluation methods that only provide arcane scores, TIGERScore is guided by the natural language instruction to provide error analysis to pinpoint the mistakes in the generated text. Our metric is based on LLaMA-2, trained on our meticulously curated instruction-tuning dataset MetricInstruct which covers 6 text generation tasks and 23 text generation datasets. The dataset consists of 48K quadruple in the form of (instruction, input, system output → error analysis). We collected the 'system outputs' through diverse channels to cover different types of errors. To quantitatively assess our metric, we evaluate its correlation with human ratings on 5 held-in datasets, 2 held-out datasets and show that TIGERScore can achieve the highest overall Spearman's correlation with human ratings across these datasets and outperforms other metrics significantly. As a reference-free metric, its correlation can even surpass the best existing reference-based metrics. To further qualitatively assess the rationale generated by our metric, we conduct human evaluation on the generated explanations and found that the explanations are 70.8% accurate. Through these experimental results, we believe TIGERScore demonstrates the possibility of building universal explainable metrics to evaluate any text generation task.

## 1 INTRODUCTION

Evaluation of natural language generation tasks is a long-standing challenging problem. With the recent advancement of large pre-trained language models like OpenAI GPT (Brown et al., 2020; OpenAI, 2023), LLaMA (Touvron et al., 2023), generative AI has become more popular than ever. Newly developed text generative models are being deployed across a wide range of downstream applications. As more and more people use these generative models, there is an increasing need to develop trustworthy evaluation metrics. However, the existing automatic metrics are lagging and mainly suffer from specific issues:

**Dependency on references**: Some evaluation metrics like ROUGE (Lin, 2004), BLEU (Papineni et al., 2002b), COMET (Rei et al., 2020), InstructScore (Xu et al., 2023c) would require gold references to measure the quality. These metrics compare the generated output against one or more reference texts to assign the evaluation score. However, such assumption can be highly unrealistic in many downstream applications where the golden reference is hard to collect.

**Limited to specific domains**: Some evaluation metrics are limited to specific domains, lacking the ability to generalize to broader text generation tasks. For example, COMET (Rei et al., 2020), BLEURT (Sellam et al., 2020), SESCORE2 (Xu et al., 2023b) and InstructScore (Xu et al., 2023b) are specifically designed for tasks like machine translation or WebNLG tasks.

**Lack of attribution:**: Some evaluation metrics tend to directly output a score without any attributions, e.g. where the errors occur and why. For instance, BARTScore (Yuan et al., 2021), BERTScore (Zhang et al., 2019) and GPTScore (Fu et al., 2023) adopt the pre-trained language models' log likelihood as the evaluation metric. Such metrics do not provide the location and reason of the assigned score, which limits their trustworthiness and reliability.

To address these issues, we propose a novel metric, TIGERSCORE, a **T**rained metric that follows **I**nstruction **G**uidance to perform **E**xplainable and **R**eference-free evaluation. As shown in Figure 1,

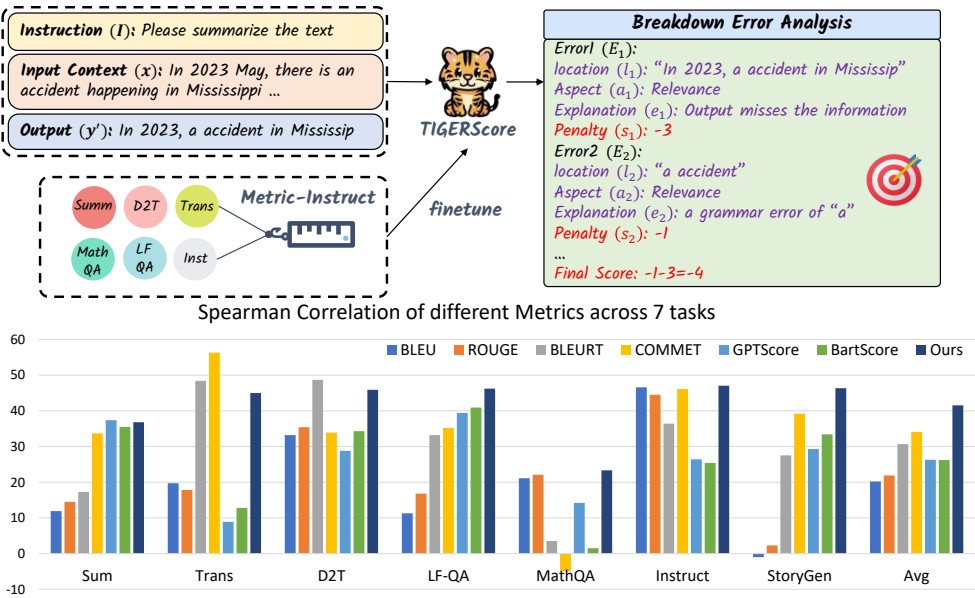

Figure 1: The upper part shows the input and output format of our metric. The lower part shows the Spearman's correlation of different metrics w.r.t human ratings.

the input to TIGERSCORE consists of an instruction describing task definition, the task input, and system output. TIGERSCORE is capable of generating breakdown error analysis that can (1) locate each mistake, (2) explain the mistake and suggest revisions, and (3) provide a penalty score (between [-5, -0.5]) for each mistake. The final score can be calculated by summing all the penalty scores.

TIGERSCORE is built by fine-tuning LLaMA-2 (Touvron et al., 2023) on our curated `MetricInstruct` dataset, which contains a total of 48K examples of (instruction, input, system output, error analysis), obtained from 23 text generation datasets. The dataset includes system outputs from more than 50 real-world systems, covering a wide variety of errors. The error analysis is generated by prompting OpenAI GPT models (OpenAI, 2023) and filtered through various heuristic-based strategies. The tuned model TIGERSCORE has shown the highest overall Spearman's correlation with human ratings on seven major text generation tasks, including summarization, translation, and long-form QA. As a reference-free metric, TIGERSCORE can even surpass the best reference-based metrics like UniEval (Zhong et al., 2022b) by 5% in terms of Spearman's correlation. We further evaluated the explanations generated by TIGERSCORE and found that over 70% of the generated explanations are accurate and trustworthy. Our analysis shows that the success of TIGERSCORE is attributed to three key aspects in `MetricInstruct`: (1) dataset diversity, (2) error coverage, and (3) high quality, which enable TIGERSCORE to generalize better than any other metric.

## 2 TIGERSCORE

TIGERSCORE is built upon three design criteria: (1) It is driven by instructions, making it easily adaptable to any text generation task. (2) The evaluation process eliminates the need for a "gold standard" or perfect example for comparison. (3) It is highly explainable, as the model can generate an error analysis that helps the user understand each identified mistake and its associated penalty.

### 2.1 BACKGROUND

The pursuit of improved metrics for text evaluation has been a significant focus since the inception of language models. Automatic n-gram-based metrics (Elliott & Keller, 2014; Callison-Burch et al., 2006; Isozaki et al., 2010) have always served as the default metric, computing the n-gram match F-1 score with the reference text until research highlighted their significant weaknesses in aligning with human preferences. Later, Neural Metrics were proposed to capture semantic similarity by either computing based on neural representation (Zhang et al., 2019; Yuan et al., 2021) or directly fine-tuning with human preferences (Rei et al., 2020). It has also been demonstrated that multi-aspect

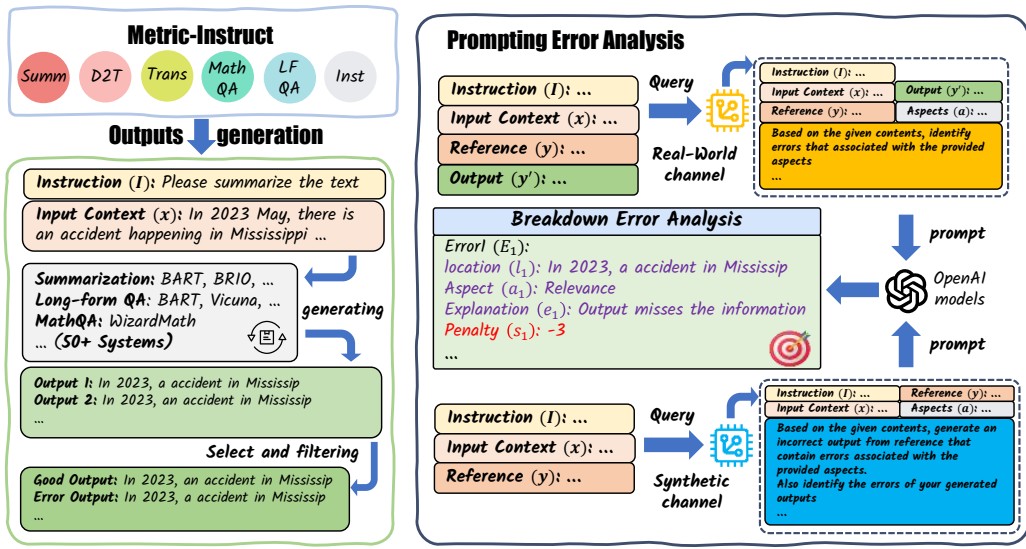

Figure 2: Overall pipeline of constructing `MetricInstruct` through the two-channel collection

scores, using the logits of large language models with well-designed instructions as prompts (Fu et al., 2023), could achieve an even higher correlation.

There have been some attempts to build explainable metrics leveraging the great capacity of large language models. For instance, UniEval (Zhong et al., 2022a) constructs a multi-aspect evaluation system by individually training on aspect-specific data. PandaLM (Wang et al., 2023a) compares two responses for a given instruction and input to judge which is better, providing a short reason for its decision. InstructScore (Xu et al., 2023c) evaluates the quality of translation by training Llama-2 to compare the reference and translation, listing errors with structured information. However, none of these metrics has been able to address all the issues mentioned in section 1 concurrently.

## 2.2 MULTI-ASPECT EVALUATION

We adopt a multi-aspect evaluation approach to provide a comprehensive view. We define unique aspects for different text generation tasks, as exemplified in Table 1. For each task, these aspects are designed to be both non-overlapping (mutually exclusive) and collectively exhaustive. Rather than explicitly highlighting these aspects and definitions in the prompt, we aim to implicitly incorporate this aspect-specific knowledge into the model during the fine-tuning stage. This approach allows the evaluation aspects to be indicated by the instruction that describes the task, encouraging the model to learn the underlying correlation between the evaluation aspects and the task definition. As a result, the model is able to generalize to unseen text generation tasks through instruction.

| Task | Aspect | Definition |
|---|---|---|
| Instruct | Comprehension | Evaluates how well the output understands the given instruction. |
| | Accuracy | Measures the correctness of the output in relation to the instruction and the paired input context. |
| | Informativeness | Assesses the relevancy and usefulness of the information provided by the output. |
| | Coherence | Evaluates how logically the output flows and connects. |

Table 1: Definitions of evaluation aspects of TIGERSCORE for Instruction-following (Instruct) as an example. See full table in Table 8 for aspect definitions of all 6 text generation tasks

## 2.3 PROBLEM FORMULATION

Suppose $y'$ is the system output from a given source context $x$ with a specific natural language instruction $I$ to describe the task definition. And $y$ is a corresponding reference output. If a metric uses $y$, it's reference-based, otherwise, it's reference-free. For example, when $T$ refers to "translation", an instruction $I$ for that task could be "Translate the following text from Chinese to English". For each task type $T$, we ask the evaluation metric to focus on a few pre-defined evaluation aspects $A_T$ like relevance, factuality, fluency, etc.

| Task | Real-World (training set) | | | Synthetic (training set) | | |
|---|---|---|---|---|---|---|
| | Dataset | Output Source | # Sample | Dataset | Output Source | # Sample |
| Summ | SummEval‡, XSum, Newsroom,SAMSum | 27 Systems | 4179 | CNN/DM,XSum, Gigaword,SAMSum | GPT-4 | 612 |
| Trans | WMT-22‡ | 18 Systems | 5070 | WMT-22 | GPT-4 | 672 |
| D2T | WebNLG-2020‡ WikiTableText,ToTTo | 17 Systems | 1603 | WikiTableText Dart,ToTTo | GPT-4 | 160 |
| LF-QA | ASQA,FeTaQA CosmosQA,ELI5 | 5 Systems | 3370 | ASQA,FeTaQA Cosmos QA,ELI5 | GPT-4 | 2146 |
| MathQA | GSM8K | 5 Systems | 4529 | N/A | N/A | 0 |
| Instruct | MixInstruct‡ | 11 Systems | 2248 | AlpacaFarm,Dolly Guanaco,OASST | GPT-4 | 3014 |

Table 2: The composition of our dataset. For synthetic data, the output is generated by asking GPT-4 to synthesize incorrect outputs that contain a few specific types of errors. For the datasets with ‡, we take their released system outputs. For the others, we collect the system outputs on our own.

TIGERSCORE is a reference-free metric, defined by a function $f$ to take the instruction $I$, source context $x$, and the system output $y'$ as input, to produce an error analysis as a list of structured errors $\{E_1, ..., E_m\}$. The error analysis is generated as follows:

$$\{..., E_i, ...\} = \{..., (l_i, a_i, e_i, s_i), ...\} = f(I, x, y') \tag{1}$$

where $(l_i, a_i, e_i, s_i)$ denotes specific information of the error $E_i$. Specifically, $l_i$ points to the location of the error, and $a_i \in A_T$ is a pre-defined aspect to which this error belongs. $e_i$ comprises both the explanation of this error and its revision suggestion. $s_i$ is the penalty score reduced for this error, which lies in $[-5, -0.5]$. The final score of $y'$ is computed as the sum of all the penalty scores: $s_{y'} = \sum_i s_i$. The range of the final score lies within $(-\infty, 0]$, where 0 means perfect generation and a lower score indicates worse quality.

## 2.4 TRAINING SETUP

TIGERSCORE is finetuned on Llama-2-7B and Llama-2-13B (Touvron et al., 2023) respectively with both's batch size being 128. The model's maximum context length is set to 1024. We use a cosine learning scheduler with the warmpup ratio being 0.1 We finetune the 7B version on 4 A100 GPUs for 3 epochs with a learning rate of 2e-5. And 13B version is run on 8 A100 GPUs for 2 epochs with a learning rate of 1e-5. Inference of TIGERSCORE is conducted on a single A100 GPU with the assistance of the VLLM toolkit to increase the speed (Kwon et al., 2023).

## 3 METRICINSTRUCT

We present the `MetricInstruct` dataset, which is employed to fine-tune TIGERSCORE. The three underlying criteria for dataset construction are: (1) **dataset diversity**: we choose 23 distinctive datasets as the source context to cover enough generation tasks. (2) **error coverage**: we take system outputs generated from 50+ text generation systems to cover all types of errors and guarantee a balanced distribution. (3) **quality ensurance**: to ensure `MetricInstruct` is tailored to gather in-depth error analysis as detailed in subsection 2.3, we sourced it by prompting OpenAI GPT models (OpenAI, 2023) and then filtered through different heuristics to eliminate low-quality error analysis.

### 3.1 DIVERSE DATASET SOURCE

`MetricInstruct` incorporates samples from 23 distinctive text generation datasets, which are categorized into 6 major categories of text generation tasks. While the collection encompasses well-researched tasks such as summarization (Summ), translation (Trans), and data2text (D2T), it also introduces popular new tasks like Long-Form QA (LF-QA), MathQA, and instruction-following (Instruct). These latter tasks have witnessed limited evaluation research. Although the assessment of traditional tasks has dominated the research landscape, we posit that effectively evaluating these new tasks is crucial for constructing a comprehensive evaluator for all text generation domains.

In addition, we meticulously selected the dataset for each task to ensure a diverse coverage across the knowledge domain. For instance, in the case of LF-QA, we utilize ASQA (Stelmakh et al., 2022b)

| Task | Eval Dataset | Output Source | # Inputs | # Samples |
|------|-------------|---------------|----------|-----------|
| Held-in Evaluation Dataset (test set) | | | | |
| Summarization | SummEval | 16 Systems | 100 | 1600 |
| Translation | WMT-22 (zh-en) | 18 Systems | 1875 | 33750 |
| Data to Text | WebNLG-2020 | 18 Systems | 179 | 2848 |
| Long-form QA | ASQA, FeTaQA, CosmosQA, ELI5 | 4 Systems | 400 | 1600 |
| Math QA | GSM8K | 2 Systems | 1319 | 2638 |
| Held-out Evaluation Dataset (test set) | | | | |
| Instruct | LIMA,AlpacaEval | 9 Systems | 500 | 4500 |
| Story Generation | OpenMEVA (ROC) | 5 Systems | 200 | 1000 |

Table 3: Overview of our main evaluation datasets. "# Samples" is computed by multipling "# Inputs" with the number of outputs systems, representing the total instances that a metric need to compute. Performance of baselines and TIGERSCORE on these datasets are reported in Table 4.

to address ambiguous factoid questions, while FeTaQA (Nan et al., 2022b) is employed to handle challenges related to tabular source question answering. We then sampled portions of each dataset's training set, guided by a series of predefined constraints such as maximum input/output length.

## 3.2 BROAD COVERAGE OF ERRORS

As shown in Figure 2, our 'system outputs' come from two channels, namely real-world system outputs and synthetic outputs. The real-world system outputs are obtained from real systems, which ensures the error distribution is aligned with real-world ones. However, the real-world systems can overfit to certain errors while missing the others. Therefore, we synthesize model outputs that can balance under-represented errors by prompting GPT models (OpenAI, 2023).

**Real-World:** We consider a wide range of systems and use their outputs as our evaluation input. Some of the system outputs are collected from the existing work like SumEval (Fabbri et al., 2021), while the others are collected by us by prompting existing domain-specialized models. To assess the quality of the output in a structured manner, we use BARTScore as our evaluation metric. We categorized the outputs into three groups: the top one-third with the highest BARTScore, the middle one-third with intermediate scores, and the bottom one-third with the lowest scores. For Math QA, the output is determined by selecting one from the five candidate options. The intention is to select more outputs that might contain errors as some systems rarely make mistakes.

Subsequently, we utilized carefully designed prompting templates (see in A.3) to elicit standardized error analysis from OpenAI models, specifically GPT-4. The main idea is to provide them with the instruction $I$, input $x$, the reference output $y$, and system output $y^I$ along with the definitions of pre-defined aspects $A_T$, to query the OpenAI models to generate $E_i$, as described in subsection 2.3. We also report their correlation performance in Table 4, showcasing the reference-based ChatGPT results.

**Synthetic:** To complement the real-world system outputs to cover broader error aspects, we instructed GPT-4 to deliberately generate designated erroneous outputs, modified from the existing reference output $y$. By supplying GPT-4 with a combination of randomly selected aspects and their definitions $A_T^I$, we control the aspect of errors it produces so that error aspects can be more balanced. Consequently, GPT-4 produced the synthesized incorrect output accompanied by explanations $E_i$, which then served as labels for subsequent fine-tuning.

## 3.3 HEURSTIC-BASED FILTERING

Upon receiving preliminary results from the **real-world** and **synthetic** channels, we further post-process these results to derive the final `MetricInstruct`. To ensure data quality and reliability, we implement a series of steps. Initially, we filter out any anomalous data in JSON format. Then, we utilize a set of rules to eliminate data with unclear format or logical issues, for instance, the error analysis is deemed unreasonably scored if the score reduction exceeds 2.5 for an error with a severity of Minor. To give the reference-free nature of TIGERSCORE, data relying on reference outputs for justification are excluded. This is crucial since explanations mentioning the unspecified reference during finetuning would make the model learn unwarranted inferences. In advance, to mitigate the

impact of the hallucination in error analysis, we split the error location and the source by spaces. Then we remove error positions that contain too many words that do not appear in the input source to avoid the illusion of error locations. Finally, after a series of manual analyses and trivial preliminary experiments, we can determine the optimal mix ratio and dataset size shown in Table 2, to ensure that the data distribution between these two channels is comparable and can serve as a complementary source. This mix ratio plays a vital role in determining the composition of the final `MetricInstruct` and training our TIGERScore.

# 4 EXPERIMENTS

## 4.1 EVALUATION DATASETS

We have gathered both the held-in and held-out datasets to compare the performance of TIGER-SCORE with the existing baseline metrics. The basic statistics of some main datasets for each task are shown in Table 3. System outputs to be evaluated of each test dataset are either from official releases, such as SummEval (Fabbri et al., 2021), WebNLG-2020 (Zhou & Lampouras, 2020), WMT-22 (Freitag et al., 2022), and OpenMEVA (Guan et al., 2021), or by generating by ourselves, such as A-F-E-C (Stelmakh et al., 2022a; Nan et al., 2022a; Fan et al., 2019; Huang et al., 2019), GSM8K (Cobbe et al., 2021), LIMA (Zhou et al., 2023) and AlpacaEval (Li et al., 2023).

Human preference scores are necessary to conduct the correlation analysis. Those datasets with official system outputs released are usually accompanied by systematic human preference scores, like the WMT-MQM score for the translation task. However, these scores are not available for tasks like long-form QA, MathQA, and instruction-following.

Therefore, we here introduce what human preference scores we have used to conduct the evaluation experiments. For summarization, data2text, and story generation tasks, we use their official human ratings from multiple aspects of their released outputs. For translation, we use the official WMT-22 MQM scores as the gold scores. For MathQA, we simply use the accuracy (1 or 0) as the gold preferences. For instruct-following, we use human ratings from the hugging face community of a dataset where 500 instances are sampled from LIMA and AlpacaEvla. For Long-form QA, we use the powerful GPT-4 to perform pairwise comparisons for them and count the winning times as the way to rank them, which is similar to how Jiang et al. (2023) constructs MixInstruct.

## 4.2 BASELINES

We categorize the baselines into reference-based and reference-free metrics for fair comparison.

**Reference-based:** We choose popular metrics, including BLEU (Papineni et al., 2002a), ROUGE (Lin, 2004), BERTScore (Zhang et al., 2019), BLEURT (Sellam et al., 2020) and BARTScore (Yuan et al., 2021). Recent emerging metrics are also included, like COMET-22 (Rei et al., 2022a),UniEval (Zhong et al., 2022b), and GPTScore (Fu et al., 2023). Specifically, we use BARTScore-ref to denote that we adopt the ref-hypo scoring type. For GPTScore, we use FLAN-T5-base (Chung et al., 2022) as base models and use GPTScore-ref to denote that we adopt the f-1 average of the ref-hypo and hypo-ref scores. Besides, we also report the correlation performance of directly prompting ChatGPT (GPT-3.5-turbo) with the prompting templates specifically designed for each task in section 3.2. Reporting this baseline will help us understand whether TIGERSCORE has surpassed its teacher model after being finetuned on `MetricInstruct`.

**Reference-free:** We choose BARTScore, GPTScore and COMETKiwi (Rei et al., 2022b) as reference-free baselines to compare. Specifically, we use BARTScore-src to denote the src-hypo scoring type, thus making it a reference-free metric. For GPTScore, we still use the FLAN-T5-base model and use GPTScore-src to denote that we use the src-hypo scoring type.

## 4.3 MAIN RESULTS

We present a comprehensive analysis of TIGERSCORE across all 6 held-in tasks and 1 held-out task in Table 4 reporting Spearman correlation results. Additionally, we provide supplementary results on Pearson and Kendall correlations in the Appendix (see Table 18 and Table 20). We average the performance across these tasks to gauge the general ability of the model.

| Tasks→
Metrics↓ Datasets→ | Summarization
SummaEval | Translation
WMT22-zh-en | Data2Text
WebNLG2020 | Long-form QA
ASQA+ | MathQA
gsm8k | Inst-Fol
LIMA+ | Story-Gen
ROC | Average |
|---|---|---|---|---|---|---|---|---|
| *GPT-based Metrics* | | | | | | | | |
| GPT-3.5-turbo (few-shot) | **38.50** | 40.53 | 40.20 | 29.33 | **66.46** | 23.20 | 4.77 | 34.71 |
| GPT-4 (zero-shot) | 36.46 | **43.87** | 44.04 | **48.95** | 51.71 | **58.53** | 32.48 | **45.15** |
| *Reference-based Metrics* | | | | | | | | |
| BLEU | 11.98 | 19.73 | 33.29 | 11.38 | 21.12 | **46.61** | -1.17 | 20.42 |
| ROUGE-2f | 14.53 | 17.83 | 35.49 | 16.83 | 22.12 | 44.56 | 2.34 | 21.96 |
| InstructScore | 26.33 | 47.30 | 43.93 | 21.62 | -4.15 | 16.19 | 16.13 | 23.91 |
| GPTScore-ref | 14.73 | 24.95 | 39.42 | 31.60 | 18.20 | 33.14 | 18.24 | 25.75 |
| BARTScore-cnn(hypo-ref) | 13.64 | 28.53 | 36.12 | 29.57 | **23.35** | 32.49 | 26.64 | 27.19 |
| BARTScore-para (hypo-ref) | 17.18 | 33.72 | 40.79 | 28.94 | 17.27 | 34.47 | 17.43 | 27.11 |
| BERTScore | 23.67 | 42.41 | 43.75 | 25.60 | 11.53 | 45.77 | 2.88 | 27.95 |
| BLEURT | 17.30 | 48.41 | **48.76** | 33.26 | 3.53 | 36.46 | 27.52 | 30.75 |
| UniEval(summ) | **47.52** | 21.90 | 38.38 | **41.83** | 19.78 | 16.02 | **44.46** | 32.84 |
| COMET-22 | 33.75 | **56.35** | 33.92 | 35.28 | -5.53 | 46.13 | 39.20 | **34.16** |
| *Reference-free Metrics* | | | | | | | | |
| BARTScore-para (src-hypo) | **38.68** | 9.60 | 32.26 | 26.86 | -2.70 | 5.92 | 20.55 | 18.74 |
| BARTScore-cnn (src-hypo) | 35.50 | 12.83 | 34.33 | 40.96 | 1.50 | 25.43 | 33.48 | 26.29 |
| Llama-2-13b-chat-0-shot | 28.53 | 14.38 | 29.24 | 19.91 | 1.08 | 21.37 | 26.78 | 20.18 |
| COMETKiwi | 16.27 | **48.48** | 27.90 | 18.05 | -11.48 | 34.86 | 18.47 | 21.79 |
| GPTScore-src | 37.41 | 8.90 | 28.82 | 39.48 | 14.25 | 26.46 | 23.91 | 25.61 |
| TIGERScore-7B-V1.2 (ours) | 35.11 | 41.50 | 42.39 | **47.11** | 21.23 | 43.57 | 39.26 | 38.60 |
| TIGERScore-13B-V1.2 (ours) | 36.81 | 44.99 | **45.88** | 46.22 | **23.32** | 47.03 | 46.36 | **41.52** |
| Δ (ours - best reference-free) | -2 | -3 | +12 | +5 | +9 | +14 | +13 | +16 |

Table 4: The Spearman correlation results of all the baseline metrics and TIGERSCORE on the evaluation datasets shown in Table 3. For each task, the metric with the highest correlation to average performance is highlighted in bold.

Our results highlight the significant advantages of TIGERSCORE over other reference-free metrics. Notably, TIGERSCORE has surpassed all other reference-free metrics in Kendall correlation. In Spearman correlation, TIGERSCORE is the highest for 5 out of 7 tasks. This underscores the robustness and consistency of TIGERSCORE in evaluating text generation tasks.

When compared with reference-based baselines, TIGERSCORE generally outperforms most reference-based metrics. However, it does score lower than some task-specific metrics like UniEval (summ) in summarization, COMET-22 in translation, and BLEURT in data2text. We consider these discrepancies acceptable, given that the compared metrics are reference-based and all specifically finetuned for a single task. Remarkably, in four other tasks, TIGERSCORE exceeds the correlation scores of all reference-based metrics, in both single-task and all-task average aspects, reaffirming its effectiveness as a universal metric.

Furthermore, we include results from GPT-3.5-Turbo (few-shot) and GPT-4 (zero-shot) in our analysis. For GPT-3.5-Turbo, we used our custom-designed templates (detailed in **??**), and for GPT-4, we simply requested a score out of 10. The findings indicate that TIGERSCORE performs comparably with GPT-4 zero-shot, especially excelling in translation, data2text, and story generation tasks.

## 4.4 HUMAN EVALUATION

The reasonableness of the error analysis provided by TIGERSCORE was assessed through a random selection of 50 error analyses from each evaluation dataset. These error analyses are then evaluated by human experts who rated them from the following perspectives:

**Reasonableness:** The human experts directly pointed out which errors are problematic in error analyses, examining whether the analysis contained hallucination or illogical reasoning.

**Comprehensiveness:** The human experts carefully review the source, output, and error analyses to determine if there are any additional errors unnoticed by TIGERSCORE. Based on human experts' analysis, they give a score on a scale of 1 to 4, specifically focused on identifying potential errors that may have been overlooked in the original analysis conducted by TIGERSCORE.

**Effectiveness:** The revision suggestions in error analyses are evaluated by human experts, on a scale of 1 to 5, to determine their appropriateness and effectiveness in enhancing the output quantity.

**Overall:** The Human experts further assign an overall score on a scale of 1 to 5 based on the reasonableness, comprehensiveness, and effectiveness of the error analysis.

Based on the results of the human evaluation in Table 5, it was found that 64.3% of TIGERSCORE's error analyses were deemed reasonable, that is, the answer to the first question is "no errors in interpretation". This suggests that most error analyses accurately identified and explained errors. In

| Aspects | Explanation Error? | | Overlooked Errors | | | | Revision Suggestions | | | | | Overall Rating | | | | |
|---|---|---|---|---|---|---|---|---|---|---|---|---|---|---|---|---|
| Rate→ | No | Yes | 1 | 2 | 3 | 4 | 1 | 2 | 3 | 4 | 5 | 1 | 2 | 3 | 4 | 5 |
| Summ | **70** | 35 | 2 | **17** | 15 | 16 | 6 | 4 | **19** | 7 | 14 | 3 | 10 | **17** | 7 | 13 |
| Trans | **54** | 25 | 3 | 8 | 17 | **22** | 2 | 7 | 17 | 6 | **18** | 3 | 6 | 15 | 9 | **17** |
| D2T | 19 | **21** | 1 | 8 | 10 | **31** | 11 | 8 | 9 | 3 | **19** | 11 | 10 | 4 | 7 | **18** |
| LF-QA | **42** | 19 | 4 | 10 | 11 | **25** | 5 | 8 | 14 | 7 | **16** | 6 | 8 | 10 | 6 | **20** |
| MathQA | **39** | 26 | 5 | 12 | 12 | **21** | 5 | 7 | **19** | 5 | 14 | 4 | 9 | 10 | 13 | **14** |
| Instruct | 5 | **9** | 5 | 5 | 8 | **32** | **21** | 3 | 5 | 2 | 19 | 9 | 4 | 3 | 7 | **27** |
| StoryGen | **66** | 29 | 7 | **16** | 13 | 14 | 7 | 6 | **16** | 10 | 11 | 7 | **12** | 11 | 9 | 11 |
| Total | **295** | 164 | 27 | 76 | 86 | **161** | 57 | 43 | 99 | 40 | **111** | 43 | 59 | 70 | 58 | **120** |

Table 5: Human evaluation results, the first question is asked per error in error analyses, and the others are per sample. Superior performance is indicated by higher numerical values. The most-voted rate of each task for each human evaluation aspect is bolded.

70.6% of cases, evaluators gave a positive score (3 or 4) for question 2, implying no missing errors were found. This demonstrates TIGERSCORE's effectiveness in comprehensive error analysis. Furthermore, in 71.4% of error analyses receiving a score of 3 to 5, TIGERSCORE's explanations contributed to text revision, guiding users towards appropriate improvements. Overall, 70.8% of error analyses received positive ratings (3 to 5), indicating good quality and usefulness in identifying and explaining errors according to human experts. Additionally, TIGERSCORE achieved the highest score (5) in 34.3% of error analyses, highlighting its high reliability and value.

## 4.5 ABALTION STUDY ON DATA SOURCE

**Ablation of Two Channel Data Collection**    To investigate the significance of our two-channel data in contributing to TIGERSCORE's strong performance, we conducted experiments with three setups: **TIGERSCORE (`MetricInstruct-Real-World`)** to assess the quality and representation of real-world data, **TIGERSCORE (`MetricInstruct-Synthetic`)** to evaluate the impact of synthetic data as a complement, and **TIGERSCORE (`MetricInstruct-Mix`)**, our official model, to assess the combined effect.  The results in Figure 3 demonstrate that our mixed-sampling training method improves most tasks. Notably, TIGERSCORE(MetricInstruct-Mix) outperforms other methods for most tasks, followed by TIGERSCORE(MetricInstruct-Real-World), validating effective representation of real-world channel data.  Although TIGER-SCORE(MetricInstruct-Mix) does not perform well on all tasks, such as data2text and story generation, it does not perform as well as TIGERSCORE(MetricInstruct-Real-World). However, it has no obvious shortcomings and has high robustness. Despite the subpar performance of TIGER-SCORE(MetricInstruct-Synthetic), the improvement between TIGERSCORE(MetricInstruct-Mix) and TIGERSCORE(MetricInstruct-Real-World) suggests valuable supplementation from synthetic data in enhancing model robustness.  These experiments unequivocally validate the indispensable role of both real-world and synthetic data in driving exceptional performance forTIGERSCORE. To be specific, the poor performance of TIGERSCORE(MetricInstruct-Real-World) on Long-Form QA can be attributed to the absence of the gold reference, which leads to a tendency of redundant errors.

**Ablation of Each Single Generation Task**    In order to investigate the contribution of each task in the `MetricInstruct`, we conducted experiments to verify whether multi-task training will gain, where we trained a series of models on individual tasks and compared its performance to that of TIGERSCORE.  As depicted in the Table 6, our unified model outperforms others for almost every task, highlighting the effectiveness of the combined tasks in achieving superior results, which

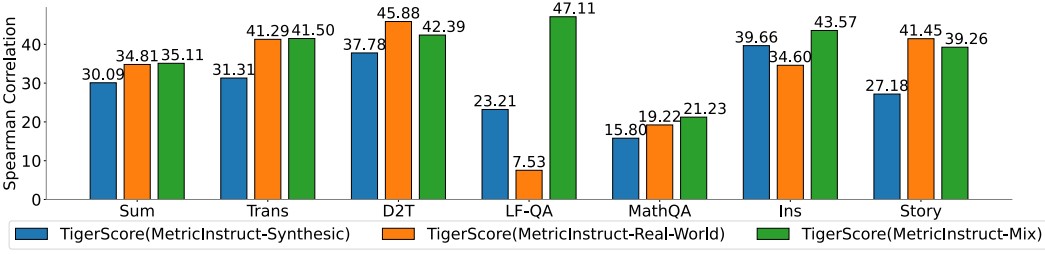

Figure 3: Investigation of the influence of Real-World & Synthesic mix training on the 7B model.

| Metrics↓ Tasks→ | Summarization | Translation | Data2Text | Long-form QA | Math | Inst-Fol | Average |
|---|---|---|---|---|---|---|---|
| Single Task | 35.55 | 41.65 | 39.75 | 41.60 | **26.75** | 41.19 | 37.75 |
| Multi Task | **36.81** | **44.99** | **45.88** | **46.22** | 23.32 | **47.03** | **40.71** |
| Δ (%) | 3.56 | 8.02 | 15.43 | 11.09 | -12.83 | 14.18 | 7.84 |

Table 6: Ablation of the influence of multiple tasks mix or single task on the 13B model.

suggests that the tasks within the `MetricInstruct` framework are not only complementary but also synergistic. However, the individual model outperformed the Math QA task. We contend that Math QA poses a significant challenge for LLMs, and a separately trained model is more adept at handling this task.

## 5 RELATED WORK

### 5.1 INSTRUCTION-DRIVEN LARGE LANGUAGE MODELS

Instruction tuning has recently become the standard to "align" language models with more useful objectives and human preferences. The instruction tuning step is normally done to enhance the certain skillset of large language models. Previously, instruction tuning has been focused on activating models' general capabilities to follow instructions to solve general tasks. Some work has been published like NaturalInstruction (Wang et al., 2022), FLAN (Wei et al., 2021) and T0 (Sanh et al., 2021) are the earliest work in the field. Later on, FLAN-v2 (Chung et al., 2022) have been proposed to understand the effect of scaling up the instruction datasets to understand its impact on model performance. These approaches mainly adopt human-annotated datasets to build the instruction following dataset. More recently, multiple works (Wang et al., 2023b; Xu et al., 2023a) propose to utilize synthetic instruction following data distilled from GPT-4 to align open-source LLMs. Our work differs from these line of work in a sense that our method aims to activate specialized capability to generate error analysis according to instruction, which is the first of its kind.

### 5.2 EXPLAINABLE METRICS

The increasing focus on model interpretability has led to a surge in research dedicated to explainable metrics. Research in these fields aim to build a metric system for a certain task that is readable to humans and is expected to help the development of better text generation systems (Leiter et al., 2022). Early endeavors in this area delved into explainability via multi-faceted evaluations, as exemplified by works such as Unieval (Zhong et al., 2022b) and GPTScore (Fu et al., 2023). With the large language model blooming, some researchers began to directly prompt LLMs to create interpretable metrics. A notable instance is the error-analysis prompting method which prompts ChatGPT to identify errors in machine translation. Some have even fine-tuned their models. One instance is PandaLM, trained on Llama to compare two responses pairwise and provide a straightforward textual rationale for its decisions (Wang et al., 2023a). Another noteworthy approach is InstructScore, leveraging large language models as knowledge repositories to obtain premium error analysis examples (Xu et al., 2023c). Despite these commendable advancements, most existing explainable metrics still require gold references and are often limited concerning the task domain. Our contribution distinguishes itself by offering a reference-free nature and the cross-task ability brought by instruction-tuning over large language models.

## 6 CONCLUSION

In this paper, we propose the novel metric TIGERSCORE, which is able to evaluate any text generation task guided by natural language instruction. We demonstrate the exceptional performance of TIGERSCORE by its high correlation with human preference. We also demonstrate the high accuracy of its generated rationale. However, TIGERSCORE does hallucinate sometimes to generate false explanation. On the other hand, we found that TIGERSCORE is not good in evaluating more open-ended text generation like story generation. In the future, we plan to devote more effort to enable more faithful explanation and unleash its potential to evaluate open-ended generation.

ETHICS STATEMENTS

Our work collected data from publicly available datasets that were ethically curated with informed consent from all participants. We ensure that all privacy data is excluded. We acknowledge the potential of our machine-learning models to generate hallucinated, biased, or unfair content. Methods have been adopted to prevent the generation of these kinds of content with our best efforts. Our research involves human evaluation experiments and we ensure that each participant's privacy is excluded and protected on our side. We make sure each participant is paid fairly according to their work amount. Our hourly rate is 12 dollars, which is above the US lowest payment rate.

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

# A  APPENDIX

## A.1  EVALUATION ASPECTS AND DEFINITIONS

With the assistance of GPT-4, we have carefully curated the evaluation aspects of each task that are mutually exclusive and collectively exhaustive. The steps include:

- Step 1: We prompt GPT-4 to output 20 candidate aspects for each task.
- Step 2: We ask GPT-4 to summarize these aspects into 3 to 5 general aspects for this task.
- Step 3: We ask GPT-4 to generate the detailed definition and 5 specific error types under each aspect.

In each step, we check the reasonability of GPT-4's response and make necessary modifications to the responses, including summarized error aspects, error definition, and error types, to make them more clear, concise and typical. After these steps, the templates in Table 8 are created and used to prompt GPT-4 to generate high-quality content.

---

You are evaluating a model-generated output for the mathematical word problem.
There might be multiple errors in the output and they can focus on different evaluation aspects.
Please list 20 evaluation aspects for mathematical word problems and their definitions.

---

Please summarize the above-provided aspects into 3 to 5
general aspects that are mutually exclusive and collectively exhaustive.

---

Mathematical word problems are a type of math task that requires model
to read and understand a text-based description of a math situation and
then formulate and solve a mathematical equation to solve the problem.
Now you are evaluating a model-generated output for a mathematical word problem.
you might want to design a few aspects where errors are identified in the outputs
could be attributed to these aspects.
These aspects could be
"Problem Understanding", "Problem Formulation",
"Computing Accuracy", "Solution Interpretation",
provide the definition of each aspect above and 5 specific error types for each aspect.
Note that these aspects are used to evaluate a single output instead of a system.

---

Table 7: An example prompt when creating the template of mathQA

| Task | Aspect | Definition |
|---|---|---|
| Summ | Relevance | The degree to which the summarized output accurately reflects the key points of the input text. |
| | Fact Consistency | If the facts in the summary are consistent with the facts in the original text. |
| | Coherence | Pertains to the logical and meaningful arrangement of ideas in the summary. |
| | Fluency | Reviews the model-generated output's use of language, including grammar, punctuation, and vocabulary that affect the quality of the sentences. |
| Trans | Accuracy | The degree to which the translated text adheres to the original text, maintaining the same meaning, context and cultural nuances. |
| | Fluency | How naturally the translation reads in the target language. |
| | Terminology | The appropriate use of specific terms and jargon related to a particular field or industry. |
| | Style Matching | Translator's ability to maintain the same style, tone, and voice as the original text. Example error types include: |
| D2T | Accuracy | Deals with the correctness of the information presented by the output. |
| | Logical Coherence | How well the output transforms structured data into a comprehensible, logical, and engaging text. |
| | Fluency | Reviews the model-generated output's use of language, including grammar, punctuation, and vocabulary that affect the quality of the sentences. |
| LF-QA | Accuracy | Evaluates the factual correctness of the answer. |
| | Completeness | Evaluates if the answer leaves out any critical parts or details that were asked in the question. |
| | Informativeness | Assesses the quality of the response in terms of how helpful it is for the user to understand the answer. |
| | Clarity | Assesses the readability and understandability of the response. |
| MathQA | Problem Understanding | Assesses how well the output accurately comprehend the text-based description of the math problem. |
| | Problem Formulation | Involves translating the problem from a textual form into a mathematical equation or set of equations that can be solved. |
| | Computing Accuracy | Assesses the output's ability to perform the mathematical operations accurately to arrive at the correct solution. |
| | Solution Interpretation | Involves the how well the output correctly interpret the solution of the problem in the context of the original problem statement. |
| Instruct | Comprehension | Evaluates how well the output understands the given instruction. |
| | Accuracy | Measures the correctness of the output in relation to the instruction and the paired input context. |
| | Informativeness | Assesses the relevancy and usefulness of the information provided by the output. |
| | Coherence | Evaluates how logically the output flows and connects. |

Table 8: Definitions of evaluation aspects of TIGERSCORE for the 6 text generation task.

## A.2 PROMPTING STRATEGIES

In our study to extract high-quality error-analysis insights from GPT-4, we employed various intuitive prompting strategies. These strategies are detailed in the prompting templates found in subsection A.3. Key strategies are outlined below:

**Two-Step Generation and Formatting Process**  One of the challenges in eliciting structured knowledge from LLMs is directing them to generate content in a specific format. Although GPT-4 often adheres to given formats, there are instances of deviations. We observed that enforcing a strict format can compromise content quality, as indicated by reduced correlation in our analysis. Our approach involves initially allowing GPT-4 to generate responses freely in the first conversation turn. In the subsequent turn, we request GPT-4 to reformat its response according to a predefined format. The initial generation templates for each task, along with a singular template for the formatting step, are listed in subsection A.3.

**Incorporation of Task-Specific Words in Initial Queries**  To leverage GPT-4's task-specific knowledge, we designed varied prompting templates for different tasks with slight modifications. Keywords like 'Source,' 'Instruction,' 'Reference,' 'Output,' 'Solution,' 'Translation,' and 'Summary' are dynamically utilized in various task contexts. In tasks like mathQA and instruction-following, where the context is self-explanatory, we omitted specific keywords.

**Integration of Predefined Aspect Definitions for Focused Evaluation**  Directly requesting GPT-4 to evaluate task outputs often led to low-quality error identification. It either pointed out simple discrepancies with the reference or missed crucial evaluation aspects, thus overlooking some errors. To address this, we incorporated predefined evaluation aspects defined in Table 8 directly into the templates, guiding GPT-4 to produce more focused responses. Exceptionally, in data2text tasks, we found that asking GPT-4 to directly evaluate errors was sufficient, and thus, we did not include our predefined aspects.

**Classification of Errors Using Major/Minor Error Guidelines**  Drawing inspiration from the MQM translation human rating system and InstructScore prompting template (Freitag et al., 2021; Xu et al., 2023c), we classified translation errors as either major or minor. This classification helped GPT-4 in assigning more consistent scores to each error, countering its instability with numerical judgments (Lu et al., 2023).

**Adopting a 0.5 to 5 Scale for Scoring Error Severity**  Initially, we used an integer scale from 1 to 5 for error penalty scores. While effective in translation tasks, this scale was less effective in tasks like summarization, data2text, and instruction-following. Our experiments demonstrated that a more nuanced scoring scale ranging from 0.5 to 5 yielded better correlation across all tasks

## A.3   PROMPTING TEMPLATES

Source: ${input_context}
Reference: ${reference_output}
Output:${hypothesis_output}
Based on the given Source and Reference, please evaluate the quality of summary(Output) written for the input text. Please score the summarization with 0.5 to 5 for aspects below. Then, identify the major and minor errors in this output for the $task task. There may be multiple errors or no error in the output. Here are the aspects you need to focus on: ${aspects_descriptions}

Table 9: Prompting templates for summarization task

Translation Instruction: ${generation_instruction}
Source Text: ${input_context}
${reference_output}
Model-generated Translation: ${hypothesis_output}
Please identify and categorize the errors in the model-generated translation as either Major or Minor. Major errors significantly impact the task, while Minor errors are subjective and represent minor imperfections. When identifying errors, do not solely rely on the reference translation for comparison. Provide explanations as an expert in the task domain, without explicitly mentioning the reference output.

Table 10: Prompting templates for translation task

Task instruction:{generation_instruction}
Source: $ {input_context}
$ {reference_output}
Output: $ {hypothesis_output}
Based on the given source and reference, identify the major and minor errors in this Output for the data to text task, which is to $generation_instruction. Note that Major errors refer to actual errors that affects the task severely, may change the meaning of the output, and Minor errors refer to smaller imperfections, and purely subjective opinions about the output. There may be multiple errors or no error in the output.

Table 11: Prompting templates for data2text task

${generation_instruction}
${input_context}
The correct solution is:
${reference_output}
A model-generated solution is: ${hypothesis_output}
Please identify all the errors in this output considering the following aspects: ${aspects_list}

Table 12: Prompting templates for mathQA task

Source: ${input_context}
${reference_output}
Output: ${hypothesis_output}
Based on the given Source and reference, identify the major and minor errors in this Output for the ${task} task, which is to ${generation_instruction}. Note that Major errors refer to actual errors that affects the task severely, may change the meaning of the output, and Minor errors refer to smaller imperfections, and purely subjective opinions about the output. You should check about ${aspects_descriptions}.There may be multiple errors or no error in the output.

Table 13: Prompting templates for long-form QA task

${generation_instruction_and_source}
${reference_output}
Output: ${hypothesis_output}
Based on the given Source and reference, identify the major and minor errors in this Output for the ${task} task. Note that Major errors refer to actual errors that affects the task severely, may change the meaning of the output, and Minor errors refer to smaller imperfections, and purely subjective opinions about the output. You should check about ${aspects_descriptions}.There may be multiple errors or no error in the output.

Table 14: Prompting templates for instruction-following task

For each error identified in your response, please provide the following information in a specific JSON format:
- Error Location: The substring in the Output that contains the error.
- Error Aspect: Choose only one from ${aspects_list}.
- Explanation: Describe why the identified issue is an error, and offer suggestions for correction. Explain as an expert in the task domain, without explicitly mentioning the reference output.
- Severity: Classify the error as "Major" or "Minor".
- Score Reduction: Assign a reduction score between 0.5 and 5, considering the severity of the error.
JSON Format for Output:
- If there are no errors:
{"errors": {}}
- If there are errors:
{"errors": {
"error_1": {
"error_location": "...",
"error_aspect": "...",
"explanation": "...",
"severity": "...",
"score_reduction": ...
},
...
}}

Table 15: Prompt template to format the response into json format

## A.4 Human evaluation case study

**Task Instruction** - Answer the following ambiguous factoid question by introducing additional knowledge, clarifying the relationship between multiple possible answers (if any) and resolving the ambiguity

**Input Context** - When did the song stayin alive come out?

**A model-generated response** - The song "Stayin' Alive" is a disco hit produced by the Bee Gees and was included in the soundtrack of the film "Saturday Night Fever" in 1977. The song was released as a single in December 16, 1577 and quickly became a popular dance anthem, topping the charts in several countries. However, it is important to note that there is another song called "Streetlife" by The Crusaders that also became popular around the same time and has a similar tempo to "Stavin'Alive", which could be confusing.

Identified errors in the model-
generated response from TigerScore-
The model-generated output contains 2 errors, with a total score reduction of 6.
Error location 1: "The song "Stayin' Alive" is a disco hit produced by the Bee Gees and was included in the soundtrack of the film "Saturday Night Fever" in 1977. The song was released as a single in December 16, 1577"
Error aspect 1: Accuracy
Explanation 1: The error is a factual inaccuracy. The song "Stayin' Alive" was not released in 1577, but in 1977. The correct information should be provided to ensure the accuracy of the answer.
Severity 1: Major
Score reduction 1: 4
Error location 2: "However, it is important to note that there is another song called "Streetlife" by The Crusaders that also became popular around the same time and has a similar tempo to "Stavin'Alive", which could be confusing."
Error aspect 2: Informativeness
Explanation 2: This is an example of irrelevance. While it is true that there is a song called "Streetlife" by The Crusaders that became popular around the same time as "Stayin' Alive", it is not relevant to the question asked. The user asked for the release date of "Stayin' Alive", not for similar songs that may have been popular at the same time. The answer should stick to the information requested.
Severity 2: Minor
Score reduction 2: 2

Table 16: A case study of our human evaluation results for long-form QA task, the main body. Corresponding to Table 17

1. Please first read through all the contents above. Which identified errors are not reasonable?

All the errors are reasonable and the identified errors are true errors

Error 1

Error 2

2. Is there any significant errors that not identified?

No. There are no missing identified errors

Yes. But the missed error is minor and does not affects the overall evaluation

Yes. And the missed error seems to a major error, but most other major errors are also identified.

Yes. And the evaluation completely misses all the major errors and does not make sense.

3. Please rate the quality of the TigerScore evaluation results for the model-generated output from 1 to 5

1. The identified errors by TigerScore does not make sense and help at all

2. TigerScore manages to identify some errors, though few explanations of them are reasonable, you can still find something useful in the outputs.

3. TigerScore is able to idenfitify part of major errors and some of explanations are helpful, despite some errors still being missed.

4. TigerScore is able to identify most of the errors some most of the explanations are helpful. The missing errors are minor and are dispensable.

5. TigerScore identified all the errors and all the explanations make sense and are very helpful.

4. Does the explantion provide helpful revision suggestions?

No. Not at all

Yes. But they are not helpful at all.

Yes. And some of them are helpful.

Yes. And most of them are helpful despite some small mistakes

Yes. And they are all helpful enough for human to revise this model-generated output.

Table 17: Human evaluations questions and corresponding answers from a human rater for the instance in Table 16

## A.5 METRICINSTRUCT DATA STATISTICS ANALYSIS

In order to find the secret of the success of MetricInstruct, we conducted a deep analysis of its inherent statistics, as shown in Figure 4. we first count the distribution of data length through the Llama tokenizer. The results show that more than 90% of data has a length lower than 1024. Due to the lack of GPU resources for fine-tuning in the longer context length scenario, this distribution further demonstrates the reasonability of using 1024 as the context length of TIGERScore.

Furthermore, we examined the incidence of errors per data instance across various tasks. The figure illustrates that each task contains most instances with the number of errors being 1 or 2 and fewer instances are considered perfect and very erroneous, reflecting a naturally occurring distribution. We contend that such a balanced distribution is crucial for the model's performance—it helps in reducing fabricated errors in correct outputs and aids in the precise identification of minor and major errors in outputs that are partially correct or entirely incorrect.

Additionally, we categorized errors as 'Major' or 'Minor' and quantified their occurrences. Our analysis reveals that tasks of a more subjective nature, such as translation and summarization, tend to have a higher frequency of minor errors. Contrastingly, in tasks like MathQA, the predominance of major errors is in alignment with the expectation that mathematical inaccuracies are generally more critical."

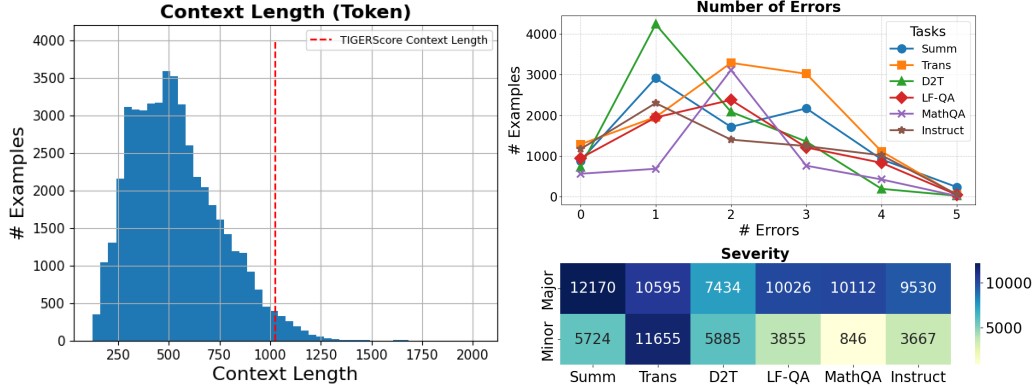

Figure 4: Distribution analysis of MetricInstruct training data for the context length, number of errors, and error severities. The left figure shows the context length distribution. The right-top plot illustrates the distribution of the per-instance number of errors. The right-bottom figure illustrates the counts of "Major" and "Minor" errors for all six tasks

### A.6 LIMITATIONS

**Hallucinated Errors**   Despite substantial efforts to minimize hallucinations in TIGERScore's output, we still observe hallucinated errors, particularly in challenging tasks such as mathQA. This issue is attributed to both the quality of our training data and the limitations of our base model. A potential solution involves initial fine-tuning on specific tasks for generation purposes, followed by further fine-tuning for evaluation. However, additional strategies are necessary to effectively reduce these hallucinations.

**Evaluation Efficiency**   TIGERScore, fine-tuned on the 7B and 13B versions of Llama-2, faces challenges with inference speed when used as an evaluation metric. Our testing reveals that TIGERScore achieves an evaluation speed of approximately 0.2 seconds per instance on a single A800 GPU with the assistance of VLLM. While this is manageable in interactive environments, further improvements are needed for efficiency in large-scale batch evaluations, compared to faster traditional metrics like BLEU and BERTScore.

**Discrepancy Between the Local Errors and Global Evaluation**   Dividing output evaluation into multiple local errors is logical, but using a simple summation of these errors as a global evaluation metric can lead to discrepancies. Longer outputs often have multiple errors, while shorter ones might be simply judged as entirely erroneous in a single error. Compared to global evaluation methods, like rating a score out of 10, developing a structured and reasonable method to accumulate and represent these errors remains an area for further exploration.

### A.7 SUPPLEMENTARY EXPERIMENT RESULTS

Our supplementary experiments mainly contain the following:

- We finished the training of TIGERScore-V1.2 using more high-quality data from GPT-4 prompting. Results are shown in Table 18, Table 19, and Table 20.
- We add more powerful baselines, including GPT4-0-shot, Llama-Chat-0-shot and InstructScore, for a comprehensive comparison.
- We include correlation metrics, Pearson and Kendall as complements for the Spearman.
- We conduct experiments that run TIGERScore on reference output for various tasks to investigate the hallucinations in the error analysis.

**TIGERScore-V1.2**    TIGERScore V1.2 adopts the same training pipeline, except that more high-quality data are used for fine-tuning. The main changes are:

1. Substituting the error-analysis data generated by ChatGPT with those produced by GPT-4.
2. Introducing 2,000 new reference evaluation examples that are error-free to enhance alignment performance.
3. Adding 10,000 instruction-following examples from alpaca, each accompanied by a GPT-4 generated free-form error analysis, to bolster generalization capabilities.
4. Employing GPT-4's evaluations to eliminate approximately 5,000 illogical instances from the training dataset.

**Alignments on no-error outputs**    In order to understand the hallucinations generated by TIGER-Score, we used TIGERScore to evaluate the gold reference, which was used in reference-based metrics, and expect TIGERScore not to hallucinate errors on these no-error instances.    Table 21.

In tasks related to instruction-following and long-form QA, TIGERScore demonstrates a high level of accuracy, avoiding hallucinations in over 85% of cases. This highlights its proficiency in producing factual and error-free analysis. However, TIGERScore is less consistent in tasks like summarization, translation, and data-to-text conversion. In these areas, it often fails to achieve perfect scores (0), but still frequently identifies gold references as either flawless or only minimally flawed (with score reductions less than 2). This could be due to the subjective nature of these tasks, where minor errors, such as the substitution of similar words, maybe more open to interpretation. Additionally, TIGERScore faces challenges in tasks like MathQA and story generation. These difficulties may stem from the inherent complexity of MathQA problems and the subjective nature of story creation, as well as specific limitations of TIGERScore in these areas. Improving TIGERScore's performance in these challenging tasks remains a topic for future research.

**Hallucinations analysis of TIGERScore outputs**    To better understand how TIGERScore handles hallucinations, we conducted experiments across six different tasks. For each task, we ran TIGER-Score on 20 samples with errors in the system output. We then used GPT-4 to determine if these samples contained hallucinations or factual inaccuracies, as outlined in the prompting templates found in Table 23. According to the results in Table 22, approximately 89.28% of TIGERSCORE's error analyses were free from hallucinations or factual errors. We acknowledge the limitations of our study, including the small sample size and the reliance on GPT-4 rather than human evaluators. Nonetheless, our findings are significant, demonstrating that TIGERSCORE is effective at avoiding hallucinations in generated content.

| Tasks→ Metrics↓ Datasets→ | Summarization SummaEval | Translation WMT22-zh-en | Data2Text WebNLG2020 | Long-form QA ASQA+ | MathQA gsm8k | Inst-Fol LIMA+ | Story-Gen ROC | Average |
|---|---|---|---|---|---|---|---|---|
| *Reference-based Metrics* | | | | | | | | |
| BLEU | 11.66 | 17.47 | 34.29 | 18.21 | 18.12 | 29.47 | -0.64 | 18.37 |
| ROUGE-2f | 16.03 | 16.26 | 35.85 | 19.66 | 20.69 | 33.49 | 2.88 | 20.69 |
| InstructScore | 27.40 | 48.49 | 46.82 | 20.59 | 0.36 | 20.98 | 12.81 | 25.35 |
| GPTScore-ref | 13.47 | 21.05 | 48.70 | 33.40 | 18.22 | 29.66 | 18.94 | 26.20 |
| BARTScore-cnn(hypo-ref) | 16.67 | 23.56 | 45.08 | 32.78 | 23.09 | 26.57 | 27.61 | 27.91 |
| BARTScore-para (hypo-ref) | 19.73 | 29.04 | 47.89 | 32.70 | 17.33 | 30.20 | 17.76 | 27.81 |
| BERTScore | 26.26 | 37.65 | 48.22 | 26.39 | 11.19 | 45.58 | 4.08 | 28.48 |
| BLEURT | 17.27 | 43.00 | **54.32** | 34.26 | 3.98 | 39.15 | 27.89 | 31.41 |
| UniEval(summ) | **53.22** | 23.11 | 51.14 | **36.95** | 17.69 | 30.87 | **44.88** | 36.84 |
| COMET-22 | 35.32 | 58.46 | 43.82 | 36.79 | -5.58 | 49.68 | 40.12 | **36.94** |
| ChatGPT-prompting | 45.53 | 43.77 | 47.76 | 29.84 | **61.26** | 15.36 | 7.80 | 35.90 |
| *Reference-free Metrics* | | | | | | | | |
| BARTScore-para (src-hypo) | 43.11 | 6.96 | 37.82 | 29.86 | -0.41 | 19.37 | 19.99 | 22.38 |
| BARTScore-cnn (src-hypo) | 39.72 | 9.53 | 45.43 | 41.48 | 3.28 | 34.97 | **33.51** | 29.70 |
| Llama-2-13b-chat-0-shot | 29.59 | 9.09 | 41.32 | 21.67 | 2.80 | 22.71 | 21.13 | 21.19 |
| COMETKiwi | 14.22 | 50.91 | 23.63 | 22.59 | -13.35 | 34.46 | 19.12 | 21.65 |
| GPTScore-src | 41.71 | 6.82 | 41.19 | 39.79 | 13.99 | 27.59 | 23.22 | 27.76 |
| TigerScore-7B-V1.0 | **45.52** | 34.52 | 50.35 | 42.45 | **33.44** | 26.97 | 29.97 | 37.60 |
| TigerScore-13B-V1.0 | 45.28 | **41.70** | 49.02 | 45.91 | 30.68 | 36.92 | 21.83 | 38.76 |
| TigerScore-7B-V1.2 | 43.95 | 37.70 | 49.13 | **46.10** | 21.77 | 38.26 | 39.90 | 39.54 |
| TigerScore-13B-V1.2 | 44.21 | 41.54 | **52.87** | 44.76 | 24.41 | 47.52 | 47.66 | **43.28** |
| GPT4-0-shot | 40.75 | 33.92 | 46.83 | 49.30 | 54.98 | 60.45 | 37.74 | 46.28 |

Table 18: The Pearson Correlation results of all the baseline metrics.

| Tasks→ Metrics↓ Datasets→ | Summarization SummaEval | Translation WMT22-zh-en | Data2Text WebNLG2020 | Long-form QA ASQA+ | MathQA gsm8k | Inst-Fol LIMA+ | Story-Gen ROC | Average |
|---|---|---|---|---|---|---|---|---|
| *Reference-based Metrics* | | | | | | | | |
| BLEU | 11.98 | 19.73 | 33.29 | 11.38 | 21.12 | **46.61** | -1.17 | 20.42 |
| ROUGE-2f | 14.53 | 17.83 | 35.49 | 16.83 | 22.12 | 44.56 | 2.34 | 21.96 |
| InstructScore | 26.33 | 47.30 | 43.93 | 21.62 | -4.15 | 16.19 | 16.13 | 23.91 |
| GPTScore-ref | 14.73 | 24.95 | 39.42 | 31.60 | 18.20 | 33.14 | 18.24 | 25.75 |
| BARTScore-cnn(hypo-ref) | 13.64 | 28.53 | 36.12 | 29.57 | 23.35 | 32.49 | 26.64 | 27.19 |
| BARTScore-para (hypo-ref) | 17.18 | 33.72 | 40.79 | 28.94 | 17.27 | 34.47 | 17.43 | 27.11 |
| BERTScore | 23.67 | 42.41 | 43.75 | 25.60 | 11.53 | 45.77 | 2.88 | 27.95 |
| BLEURT | 17.30 | 48.41 | **48.76** | 33.26 | 3.53 | 36.46 | 27.52 | 30.75 |
| UniEval(summ) | 47.52 | 21.90 | 38.38 | **41.83** | 19.78 | 16.02 | **44.46** | 32.84 |
| COMET-22 | 33.75 | 56.35 | 33.92 | 35.28 | -5.53 | 46.13 | 39.20 | 34.16 |
| ChatGPT-prompting | 38.50 | 40.53 | 40.20 | 29.33 | **66.46** | 23.20 | 4.77 | **34.71** |
| *Reference-free Metrics* | | | | | | | | |
| BARTScore-para (src-hypo) | 38.68 | 9.60 | 32.26 | 26.86 | -2.70 | 5.92 | 20.55 | 18.74 |
| BARTScore-cnn (src-hypo) | 35.50 | 12.83 | 34.33 | 40.96 | 1.50 | 25.43 | 33.48 | 26.29 |
| Llama-2-13b-chat-0-shot | 28.53 | 14.38 | 29.24 | 19.91 | 1.08 | 21.37 | 26.78 | 20.18 |
| COMETKiwi | 16.27 | **48.48** | 27.90 | 18.05 | -11.48 | 34.86 | 18.47 | 21.79 |
| GPTScore-src | 37.41 | 8.90 | 28.82 | 39.48 | 14.25 | 26.46 | 23.91 | 25.61 |
| TigerScore-7B-V1.0 | 39.32 | 42.27 | 38.42 | 42.04 | **32.07** | 25.53 | 30.22 | 35.69 |
| TigerScore-13B-V1.0 | **41.98** | **45.20** | 39.05 | 45.84 | 31.04 | 43.69 | 17.40 | 37.12 |
| TigerScore-7B-V1.2 | 35.11 | 41.50 | 42.39 | **47.11** | 21.23 | 43.57 | 39.26 | 38.60 |
| TigerScore-13B-V1.2 | 36.81 | 44.99 | **45.88** | 46.22 | 23.32 | **47.03** | 46.36 | **41.52** |
| GPT4-0-shot | 36.46 | 43.87 | 44.04 | 48.95 | 51.71 | 58.53 | 32.48 | 45.15 |

Table 19: The Spearman Correlation results of all the baseline metrics.

| Tasks→ Metrics↓ Datasets→ | Summarization SummaEval | Translation WMT22-zh-en | Data2Text WebNLG2020 | Long-form QA ASQA+ | MathQA gsm8k | Inst-Fol LIMA+ | Story-Gen ROC | Average |
|---|---|---|---|---|---|---|---|---|
| *Reference-based Metrics* | | | | | | | | |
| BLEU | 8.71 | 14.50 | 23.13 | 7.73 | 17.25 | 35.92 | -0.89 | 15.19 |
| ROUGE-2f | 10.67 | 13.19 | 24.74 | 11.73 | 18.07 | 34.59 | 1.78 | 16.40 |
| InstructScore | 20.86 | 37.69 | 32.08 | 15.64 | -3.87 | 13.87 | 13.50 | 18.54 |
| GPTScore-ref | 10.80 | 18.74 | 27.47 | 22.13 | 14.86 | 25.40 | 12.78 | 18.88 |
| BARTScore-cnn(hypo-ref) | 10.00 | 21.06 | 27.04 | 20.67 | 19.07 | 24.70 | 18.58 | 20.16 |
| BARTScore-para (hypo-ref) | 10.41 | 24.90 | 28.42 | 20.24 | 14.10 | 26.13 | 12.11 | 19.47 |
| BERTScore | 17.39 | 31.57 | 30.74 | 17.70 | 9.41 | 35.61 | 2.00 | 20.63 |
| BLEURT | 12.69 | 36.12 | **34.48** | 23.11 | 2.88 | 27.94 | 19.18 | 22.34 |
| UniEval(summ) | **35.89** | 16.08 | 28.56 | **29.32** | 16.15 | 11.93 | **31.22** | 24.17 |
| COMET-22 | 25.01 | **42.79** | 23.43 | 24.66 | -4.52 | 36.17 | 27.52 | 25.01 |
| ChatGPT-prompting | 30.45 | 32.30 | 30.38 | 20.91 | **58.57** | 17.73 | 3.26 | **27.65** |
| *Reference-free Metrics* | | | | | | | | |
| BARTScore-para (src-hypo) | 29.12 | 7.01 | 22.32 | 18.80 | -2.21 | 4.26 | 14.15 | 13.35 |
| BARTScore-cnn (src-hypo) | 26.63 | 9.40 | 23.69 | 28.93 | 1.23 | 19.09 | 23.29 | 18.89 |
| Llama-2-13b-chat-0-shot | 25.22 | 11.79 | 23.45 | 15.96 | 1.08 | 19.50 | 21.52 | 16.93 |
| COMETKiwi | 11.87 | 36.37 | 19.08 | 12.23 | -9.38 | 26.46 | 12.78 | 15.63 |
| GPTScore-src | 28.20 | 6.50 | 19.81 | 27.64 | 11.64 | 20.04 | 16.36 | 18.60 |
| TigerScore-7B-V1.0 | 32.84 | 34.42 | 30.06 | 31.56 | **30.90** | 23.24 | 24.41 | 29.63 |
| TigerScore-13B-V1.0 | **36.09** | **37.13** | 31.34 | **34.18** | 24.90 | 38.63 | 13.15 | 30.77 |
| TigerScore-7B-V1.2 | 28.79 | 33.65 | 32.44 | 33.93 | 19.98 | 38.13 | 29.72 | 30.95 |
| TigerScore-13B-V1.2 | 31.29 | 36.50 | **36.43** | 33.17 | 21.58 | **41.84** | 35.33 | **33.73** |
| GPT4-0-shot | 29.32 | 35.38 | 32.26 | 35.85 | 46.63 | 49.50 | 25.69 | 36.38 |

Table 20: The Kendall's tau Coefficient results of all the baseline metrics.

| Tasks→ | Summarization | | Translation | | Data2Text | | Long-form QA | | MathQA | | Inst-Fol | | Story-Gen | | Average | |
| Models↓Proportion of Score→ | 0 | >-2 | 0 | >-2 | 0 | >-2 | 0 | >-2 | 0 | >-2 | 0 | >-2 | 0 | >-2 | 0 | >-2 |
| TigerScore-7B-v1.2 | 16.00 | 68.00 | 3.57 | 72.48 | 45.51 | 78.65 | 84.75 | 84.75 | 34.36 | 34.36 | 73.98 | 92.48 | 34.00 | 34.00 | 41.74 | 66.39 |
| TigerScore-13B-v1.2 | 48.00 | 97.00 | 21.01 | 83.63 | 23.03 | 94.94 | 94.50 | 94.50 | 25.28 | 25.28 | 86.38 | 96.14 | 46.00 | 49.00 | 49.17 | 77.21 |

Table 21: TIGERScore score on the gold reference of the test set. For each task, the 0 column refers to the percentage that TIGERScore reduces 0 scores for the gold references. The $> -2$ column refers to the percentage that TIGERScore reduces less or equal to 1 score on the gold references.

| Tasks | Summarization | Translation | Data2Text | Long-form QA | MathQA | Inst-Fol | Story-Gen | Total |
|---|---|---|---|---|---|---|---|---|
| Accuracy | 95.00 | 95.00 | 90.00 | 85.00 | 90.00 | 75.00 | 95.00 | 89.28 |

Table 22: The accuracy of the error analysis assessed by GPT-4 that do not contain hallucinations or factual errors.if includes hallucinations or factual errors, assessed by GPT4.

Task instruction: ${instruction}
Source: ${input}
Reference output: ${reference_output}
Model-generated output:
${output}
An error analysis provided:
${error_analysis}
Does error analysis include factual errors or hallucinations of the model-generated output?

Table 23: The prompt template we used to identify whether there are hallucinated contents or factor errors in TIGERSCORE's response

