# OpenReview forum: "TIGERScore: Building Explainable Metric for All Text Generation Task"
_ICLR.cc/2024/Conference — Submitted to ICLR 2024_

### Official Review · Reviewer_A6PK · 2023-10-29

**Soundness:** 3 good
**Presentation:** 3 good
**Contribution:** 3 good
**Rating:** 6
**Confidence:** 4

**Summary:**

This paper presents TIGERScore, a Trained metric that follow Instruction Guidance to perform Explainable, and Reference-free evaluation over a wide spectrum of text generation tasks. TIGERScore is guided by the natural language instruction to provide error analysis to pinpoint the mistakes in the generated text.

**Strengths:**

*This paper proposes a novel metric TIGERScore that can provide error analysis and explanations without needing reference texts, making it widely applicable.
* Achieves state-of-the-art correlation with human judgments across multiple text generation tasks like summarization, translation, QA etc. This demonstrates its reliability.
* The error analysis and explanations make the metric more transparent and trustworthy. The human evaluation results validate their accuracy.

**Weaknesses:**

* There could be more details provided on the prompt engineering strategies used to create MetricInstruct.
* The generalizability to open-ended generation like storytelling seems limited currently. More investigation into enhancing this is needed.
* The correlation gains over prior metrics, while substantial, are incremental in some cases. The gains do not feel dramatically superior.
* There is no comparison of computational efficiency with other metrics. Efficiency can be a practical concern.

**Questions:**

NA

---

> ### Author Response · Authors · 2023-11-22
> **Response to Reviwer A6PK**
>
> Dear Reviewer `A6PK`:
>
> Thank you for your insightful review. We are glad to hear that you appreciate TIGERScore as a transparent and trustworthy metric and achieve state-of-art correlation metrics with human judgments
>
> According to the feedback of all the reviewers, we have added the corresponding experiments in the [**revised paper**](https://openreview.net/pdf?id=SIojR1ruNQ) with supplementary experiments in the appendix. We have summarized the major updates in  `General Response` (shown in a separate comment on this page).
>
> ***We would like to address your concerns in detail below.***

---

> ### Author Response · Authors · 2023-11-22
> **Response to Weakness 1 about prompt engineering strategies**
>
> > There could be more details provided on the prompt engineering strategies used to create MetricInstruct.
>
> Thank you for your suggestion to discuss prompt engineering strategies in greater detail. We agree that sharing this information can assist other researchers in crafting more effective prompt templates for their specific use cases. Consequently, we have thoroughly discussed various strategies for eliciting knowledge from GPT-4. Here, we provide a summary of these prompting strategies:
>
> When creating evaluation aspects of each task (details in appendix A.1) :
>
> - We use GPT-4 to generate candidate aspects, and then select a few of them as the final aspects with human efforts.
> - We prompt GPT-4 to generate concise definitions and typical specific error types under each aspect, and then revise them with human efforts.
>
> When designing prompting templates to get the error analysis from GPT-4 for each task, we adopt strategies (**details in appendix A.2**):
>
> - Two-Step Generation and Formatting Process
> - Incorporation of Task-Specific Words in Initial Queries
> - Integration of Predefined Aspect Definitions for Focused Evaluation
> - Classification of Errors Using Major/Minor Error Guidelines
> - Adopting a 0.5 to 5 Scale for Scoring Error Severity

---

> ### Author Response · Authors · 2023-11-22
> **Response to Weakness 2 about generalization ability**
>
> > The generalizability to open-ended generation like storytelling seems limited currently. More investigation into enhancing this is needed.
>
> Thanks for pointing out the TIGERScore V1’s problem of generalization ability. We also noticed these weaknesses after the V1 development and have been investigating the reason behind them.
>
> Our investigation shows that the low scores on story generation are caused by the lack of diversity in the dataset. Though the MetricInstruct V1 has included various text generation tasks and broad error types in training instances, they are still limited to a specific task.
>
> To increase the generation ability of TIGERScore, we collect 10k more instruction-following data from alpaca-52k and apply the same data preparation pipeline querying GPT-4 to get high-quality error analysis responses. We train TIGERScore-V1.2 on them and the results show that the generalization ability has been increased greatly (see results in Appendix A.7 )

---

> ### Author Response · Authors · 2023-11-22
> **Response to Weakness 3 about correlation performance**
>
> > The correlation gains over prior metrics, while substantial, are incremental in some cases. The gains do not feel dramatically superior.
>
> Thanks for pointing out your concern about the incremental gains. We understand your concern and summarize the following points in response:
>
> - As a reference-free metric, it’s naturally hard to achieve similar correlation performance with reference-based metrics, especially in cases where references are of high quality. We contend that one of the values of TIGERScore is its substantial gains compared to other reference-based metrics.
> - As shown in our TIGERScore V2 results, the correlation has surpassed all other reference-free metrics and achieved approaching performance with the GPT-4-0-shot. One exception is COMETKiwi, which has been specifically trained for translation on human eval rating. Therefore, it’s acceptable that TIGERScore as a more general evaluation metric fails to surpass it.
> - Except for correlation, one significant feature of TIGERScore is the structured error analyses and explanations it outputs, which most other metrics cannot provide.
>
> Therefore, we conclude that our current TIGERScore correlation has made great improvement on correlations compared to existing reference-free metrics and approached the performance of GPT-4-0-shot. And contrast to most other metrics, TIGERScore is explainable.
>
> We hope our response has solved our concern!

---

> ### Author Response · Authors · 2023-11-22
> **Response to Weakness 4 about computation efficiency**
>
> > There is no comparison of computational efficiency with other metrics. Efficiency can be a practical concern.
>
> Thank you for highlighting the importance of computational efficiency. We admit that evaluation efficiency is vital, particularly when evaluating large-scale datasets where speed determines user preference.
>
> TIGERScore, being a 7B Llama-based Large Language Model (LLM), does face challenges with inference speed. However, with the aid of the VLLM toolkit on a single A800 GPU, TIGERScore efficiently processes a single instance in approximately 0.2 seconds.
>
> In comparison, traditional metrics, such as BLUE, BERTScore, etc. are quicker, ranging from 0.1 to 0.001 seconds per instance. Nonetheless, other explainable metrics with large model sizes, like InstructScore and PandaLM. also encounter speed constraints. Traditional metrics are faster because their scores are unexplainable. TIGERScore, as an explainable metric, provides not only the score with higher correlations but also the structured error analysis content.
>
> Therefore, we believe that TIGERScore's processing time of 0.2 seconds per instance aligns well with the current industry requirements, especially in scenarios where interpretability is of great significance.

---

### Official Review · Reviewer_Az3X · 2023-11-01

**Soundness:** 3 good
**Presentation:** 3 good
**Contribution:** 2 fair
**Rating:** 5
**Confidence:** 4

**Summary:**

This paper introduces TIGERScore (Trained Instruction Guidance metric for Explainable and Reference-free evaluation) based on LLaMA 2 for NLG evaluation. For training this metric using natural language instructions, the MetricInstruct dataset is introduced, containing 48k examples and covering six text generation tasks (Summarization, Translation, Data-to-text, LongFormQA, MathQA and Instruction Following) across 23 datasets. The proposed metric provides error analysis of the generated text by 1) Locating the mistakes 2) Explaining the mistakes 3) Providing a penalty score for each of the mistakes which are then summed up to get the final score (0 meaning perfect generation). The metric is shown to empirically correlate well with human judgements.

**Strengths:**

- NLG evaluation is an interesting problem for the community which is the main focus of this paper.
- The introduced MetricInstruct dataset would be interesting for the community to develop similar novel metrics for NLG evaluation.
- Different data collection strategies, incorporating both real-world and synthetic data enhances the model's robustness.
- TIGERScore provides error analyses, and human evaluation indicates that these analyses are reasonable, comprehensive, and effective in guiding users to improve text quality.

**Weaknesses:**

- Even though the results are promising at the initial level, the spearman correlation of TIGERScore is lower than the other existing metrics for 4 out of the 7 tasks (Fig 1). Table 4 also shows a huge difference between the reference-free and reference-based metrics which hinders its usage as a universal metric (main contribution of the paper).
- The paper does not extensively explore scenarios where TIGERScore might fail to provide accurate error analyses. Understanding its limitations could be valuable for users. The paper briefly mentions the hardware used for fine-tuning but lacks an in-depth discussion of computational resources required for implementing TIGERScore.
- It would be nice to have a comparison with 0-shot GPT-4 based evaluation (having limited computational complexity) which has also been recently used as a reference free and explainable metric [1,2].
- There are no results related to hallucination or factual correctness of the generated text, which would have made the results more compelling.

[1] QLoRA: Efficient Fine Tuning of Quantized LLMs
[2] Judging LLM-as-a-Judge with MT-Bench and Chatbot Arena

**Questions:**

- Have the authors also experimented with InstructScore which seems to be very related to the proposed approach?
- Have the authors also computed inter-annotator agreement when performing human evaluation?
- Did the authors experiment with 0-shot abilities of the LLaMA-2 (both chat and non-chat) models as a reference-free evaluation using prompts?
- It is not clear why the model's max context length is set to 1024 when it can ingest 4096 tokens.
- Table 6 - do authors have any intuition why there is decreased performance for Data2Text when all tasks are included?
- Fig 3 and Table 6 - is there any specific reason why the results on MathQA and StoryGen are excluded? It would be nice to place the Figure and Table together for better comprehension.
- Is there any curriculum used for training on the different tasks?
- Would the corresponding code be made available which is necessary for reproducibility.


Suggestions/Comments:
Fig 1 caption: shows te -> shows the
Section 5.2: Research on this fields -> in these fields
Please take care of the quotes in Section 2, Table 3 caption
Please specify and distinguish between LLaMA and LLaMA2 in the abstract and the main paper.

---

> ### Author Response · Authors · 2023-11-22
> **Response to Reviewer Az3X**
>
> Thank you for your insightful review. We are delighted to hear that you recognize TIGERScore's significance on NLG evaluation and the value of MetricInstruct.
>
> According to the feedback of all the reviewers, we have added the corresponding experiments in the [**revised paper**](https://openreview.net/pdf?id=SIojR1ruNQ) with supplementary experiments in the appendix. We have summarized the major updates in  `General Response` (shown in a separate comment on this page).
>
> ***We would like to address your concerns in detail below.***

---

> ### Author Response · Authors · 2023-11-22
> **Response to Weakness 1 about correlation performance**
>
> > Even though the results are promising at the initial level, the spearman correlation of TIGERScore is lower than the other existing metrics for 4 out of the 7 tasks (Fig 1). Table 4 also shows a huge difference between the reference-free and reference-based metrics which hinders its usage as a universal metric (main contribution of the paper).
>
> Thank you for pointing out the potential weaknesses and we totally understand your concern.
>
> We first talk about the correlation performance. Our newly delivered TIGERScore-13b-V1.2 reaches the best compared to all the existing reference-free metrics on 5 out of 7 tasks, except that for summarization and translation. However, **TIGERScore maintains small gaps to the highest correlation scores**. For example, in the translation task, TIGERScore-13b-V1.2 reaches 44.99, only 3.99 to the best one in translation.
>
> Besides, we contend that Spearman should not be the only correlation metric to focus on, as suggested by reviewer `2ZJr`.  **On Kendall metric (see Appendix A.7 Table 20), TIGERScore-13b-V1.2 reaches the best across all the tasks compared to all other reference-free metrics, even surpassing GPT-4-0-shot on 4 out of 7 tasks**. Therefore, we contend that the current correlation performance of TIGERScore is super competitive.
>
> For your second concern on the gap between reference-based metrics, we have raised 3 important key features in section 1 that universal evaluation metrics should try to tackle, which are 1) Dependency on references, 2) Limited to specific domains, and 3) Lack of attribution. Compared to existing reference-based metrics:
>
> 1. **The reference feature of TIGERScore makes it easier to quickly apply to multiple scenarios, without first finding a reference.**
> 2. **The reference-based feature achieves low overall correlation performance across multiple tasks**. Considering Spearman, the best traditional reference-based metrics get 34.16 in average performance, while TIGERScore reaches 38.60 (7b) and 41.52 (13b) respectively, which is of superior advantage.
> 3. **TIGERScore is explainable through the generated error analysis, while few of the reference-based metrics are explainable (only InstructScore)**
>
> Therefore, considering all the above factors as universal metrics, we believe TIGERScore is a more qualified one than the reference-based metrics.

---

> ### Author Response · Authors · 2023-11-22
> **Response to Weakness 2 about adding limitation discussion**
>
> > The paper does not extensively explore scenarios where TIGERScore might fail to provide accurate error analyses. Understanding its limitations could be valuable for users. The paper briefly mentions the hardware used for fine-tuning but lacks an in-depth discussion of the computational resources required for implementing TIGERScore.
>
> Thank you for your constructive suggestion.
>
> We acknowledge the importance of addressing the study's limitations and have added a new section in Appendix A.6 to discuss the limitations of TIGERScore. This section covers issues in the following:
>
> 1. There are occasional generation of hallucinated errors by TIGERScore, and a need for further verification of the quality of its generated explanations.
> 2. We recognize that despite TIGERScore's ability to evaluate an instance in 0.2 seconds using VLLM, there remains a performance gap when compared to traditional metrics like BLEU and BERTScore.
> 3. We explore the discrepancies between local errors and global evaluation, concluding that more refined methods of accumulation are needed.
>
> Regarding the computational resources required for TIGERScore, **we provide a detailed discussion in Section 2.4**. This includes the use of 8 A100 GPUs for training TIGERScore-13b and 4 A100 GPUs for TIGERScore-7b. We also note that TIGERScore can be operated on a single A100 or A6000 GPU, again with the aid of VLLM, achieving an inference speed of approximately 0.2 seconds per instance. We hope that these additions adequately address your concerns.

---

> ### Author Response · Authors · 2023-11-22
> **Response to Weakness 3 and Q1 and Q3 about adding baselines**
>
> > Weakness 3: It would be nice to have a comparison with 0-shot GPT-4 based evaluation (having limited computational complexity) which has also been recently used as a reference free and explainable metric [1,2].
>
> > Question 1: Have the authors also experimented with InstructScore which seems to be very related to the proposed approach?
>
> > Question 3: Did the authors experiment with 0-shot abilities of the LLaMA-2 (both chat and non-chat) models as a reference-free evaluation using prompts?
>
> Thank you for your valuable suggestion. We agree that incorporating more robust baseline comparisons provides a clearer understanding of TIGERScore's strengths and limitations. Consequently, we have included 0-shot GPT-4 and 0-shot LLaMA2-Chat as potent reference-free baselines in Table 4. During our evaluation of the LLaMA-2-non-chat variant, we observed its inability to follow instructions to output valid scores, thus leading to its exclusion from our baseline analysis. Additionally, InstructScore is added as a powerful baseline a reference-based baseline.
>
> Our findings reveal that TIGERScore **outperforms LLaMA-2-Chat across all tasks** and delivers **comparable results to GPT-4**, demonstrating its impressive capabilities. Furthermore, TIGERScore exceeds InstructScore in all tasks except for translation, where the performance gap of translation is minimal. This slight disparity in translation is justifiable, considering InstructScore is a reference-based metric, whereas TIGERScore operates as a reference-free metric.
>
> |            Tasks$\rightarrow$             | Summarization | Translation | Data2Text  | Long-form QA |  MathQA   | Inst-Fol  | Story-Gen |  Average  |
> | :---------------------------------------: | :-----------: | :---------: | :--------: | :----------: | :-------: | :-------: | :-------: | :-------: |
> | Metrics$\downarrow$ Datasets$\rightarrow$ |   SummaEval   | WMT22-zh-en | WebNLG2020 |    ASQA+     |   gsm8k   |   LIMA+   |    ROC    |           |
> |          Reference-based Metrics          |               |             |            |              |           |           |           |           |
> |               InstructScore               |     26.33     |  **47.30**  |   43.93    |    21.62     |   -4.15   |   16.19   |   16.13   |   23.91   |
> |          Reference-free Metrics           |               |             |            |              |           |           |           |           |
> |          LLaMA-2-13b-chat-0shot           |     28.53     |    14.38    |   29.24    |    19.91     |   1.08    |   21.37   |   26.78   |   20.18   |
> |                GPT4-0-shot                |     36.46     |    43.87    |   44.04    |  **48.95**   | **51.71** | **58.53** |   32.48   | **45.15** |
> |            TigerScore-7B-V1.2             |     35.11     |    41.50    |   42.39    |    47.11     |   21.23   |   43.57   |   39.26   |   38.60   |
> |            TigerScore-13B-V1.2            |   **36.81**   |    44.99    | **45.88**  |    46.22     |   23.32   |   47.03   | **46.36** |   41.52   |
>
> A worth-noting gap between GPT-4-0-shot performance is on mathQA. We believe this severe gap is caused by 1) lack of math-specific fine-tuning, 2) inherent hardness of the mathQA task. Obviously, further work is to be explored on mathQA evaluation.

---

> ### Author Response · Authors · 2023-11-22
> **Response to Weakness 4 about hallucination analysis**
>
> > There are no results related to hallucination or factual correctness of the generated text, which would have made the results more compelling
>
> Thanks for your construction suggestion. In response to your concern, we have conducted two experiments to investigate the hallucinations of the generated text.
>
> Firstly, as suggested by reviewer `WhZB`, we run TIGERScore to evaluate the gold reference for each task to see whether TIGERScore will hallucinate errors on non-error outputs. Results are shown in the following table. Here for each task, the $0$ column refers to the percentage that TIGERScore reduces 0 scores for the gold references. The $> -2$ column refers to the percentage that TIGERScore reduces less to 2 score on the gold references.
>
>
> | Tasks$\rightarrow$                                 | Summarization |       | Translation |       | Data2Text |       | Long-form QA |       | MathQA |       | Inst-Fol |       | Story-Gen |       | Average |           |
> | -------------------------------------------------- | :-----------: | :---: | :---------: | :---: | :-------: | :---: | :----------: | :---: | :----: | :---: | :------: | :---: | :-------: | :---: | :-----: | :-------: |
> | Models$\downarrow$Proportion of Score$\rightarrow$ |       0       |  >-2  |      0      |  >-2  |     0     |  >-2  |      0       |  >-2  |   0    |  >-2  |    0     |  >-2  |     0     |  >-2  |    0    |    >-2    |
> | TigerScore-7B-v1.2                                 |     16.00     | 68.00 |    3.57     | 72.48 |   45.51   | 78.65 |    84.75     | 84.75 | 34.36  | 34.36 |  73.98   | 92.48 |   34.00   | 34.00 |  41.74  | **66.39** |
> | TigerScore-13B-v1.2                                |     48.00     | 97.00 |    21.01    | 83.63 |   23.03   | 94.94 |    94.50     | 94.50 | 25.28  | 25.28 |  86.38   | 96.14 |   46.00   | 49.00 |  49.17  | **77.21** |
>
> On average, TIGERScore (13b) does not produce any errors in the gold reference sentence about **77.21%** of the time. In these instances, it either makes no errors or only minor ones, resulting in a score reduction of less than 2.0 (minor error). **Therefore, while TIGERScore may occasionally hallucinate errors, its overall rate of hallucination is deemed acceptable.**
>
> Secondly, for each task, we ran TIGERScore on 20 samples with errors in the system output. We then used GPT-4 to determine if these samples contained hallucinations or factual inaccuracies (see Appendix A.7).
>
> | Tasks    | Summarization | Translation | Data2Text | Long-form QA | MathQA | Inst-Fol | Story-Gen | Total |
> | -------- | :-----------: | :---------: | :-------: | :----------: | :----: | :------: | :-------: | ----- |
> | Accuracy |     95.00     |    95.00    |   90.00   |    85.00     | 90.00  |  75.00   |   95.00   | 89.28 |
>
> According to the results above, approximately 89.28% of TIGERScore's error analyses were free from hallucinations or factual errors. We acknowledge the limitations of our study, including the small sample size and the reliance on GPT-4 rather than human evaluators. **Nonetheless, our findings are significant, demonstrating that TIGERScore is effective at avoiding hallucinations in generated content.**
>
> We hope our response answers your concern.

---

> ### Author Response · Authors · 2023-11-22
> **Response to Question 2 about human eval**
>
> Thank you for your suggestion. Due to budget constraints, each item undergoes only one round of human evaluation, and as a result, we do not record inter-annotator agreements.
>
> Our evaluator is selected from Prolific, a well-known crowdsourcing platform. To become an official annotator on Prolific, individuals must successfully pass specific tests. We periodically sample and review these annotations. Any poor-quality annotations, along with all other work by the responsible annotators, are excluded. This approach helps us maintain the quality of our evaluations.
>
> In summary, we are confident in the validity of our human evaluation results. **This confidence stems from the high quality of our annotators and the rigor of our review process.** We hope this explanation addresses your concerns.

---

> ### Author Response · Authors · 2023-11-22
> **Response to Question 4 about context length**
>
> > It is not clear why the model's max context length is set to 1024 when it can ingest 4096 tokens.
>
> Thank you for your insightful question!
>
> **Our decision to set the context length at 1024 is primarily driven by hardware limitations**. Currently, we lack the GPU resources necessary to fine-tune a Llama Model with full-context length, as this task demands more than 8 A100 GPUs.
>
> Additionally, our analysis of the dataset, detailed in Appendix A.5, reveals a significant finding: over 90% of the training instances have sequence lengths below 1024, with nearly all falling under 1250. This data strongly supports our choice of a 1024 context length for both fine-tuning and testing purposes.
>
> We hope this explanation clarifies your query and proves helpful.

---

> ### Author Response · Authors · 2023-11-22
> **Response to Q5 and Q6 about ablation study**
>
> > Q5: Table 6 - do authors have any intuition why there is decreased performance for Data2Text when all tasks are included?
>
> Thank you for your insightful question!
>
> Firstly, for the decreased performance for data2text when all tasks are included in TIGERScore V1, we tend to explain it that:
>
> - Data2text differs from other tasks: while other tasks utilize natural language, the source inputs of data2text are mostly structured knowledge tuples. Therefore, it might benefit little from the other knowledge through multi-task learning
> - Multi-task learning usually exhibits some extent of performance tradeoff across different tasks. And we think data2text happens to be the victim.
>
> Secondly, we want to clarify that we have conducted a brand-new ablation study based on our TIGERScore V1.2 dataset (see Table 6 or the following table), and results show all tasks except mathQA benefit a lot from multi-task learning. We blame the decrease of mathQA after multi-task learning for its inherent difficulties.
>
> | Metrics$\downarrow$ Tasks$\rightarrow$ | Summarization | Translation | Data2Text | Long-form QA |    Math   |  Inst-Fol |  Average  |
> |:--------------------------------------:|:-------------:|:-----------:|:---------:|:------------:|:---------:|:---------:|:---------:|
> |               Single Task              |     35.55     |    41.65    |   39.75   |     41.60    | **26.75** |   41.19   |   37.75   |
> |               Multi Task               |   **36.81**   |  **44.99**  | **45.88** |   **46.22**  |   23.32   | **47.03** | **40.71** |
> |              $\Delta$ (\%)             |      3.56     |     8.02    |   15.43   |     11.09    |   -12.83  |   14.18   |    7.84   |
>
> > Q6: Fig 3 and Table 6 - is there any specific reason why the results on MathQA and StoryGen are excluded? It would be nice to place the Figure and Table together for better comprehension.
>
> For mathQA, we didn't finish training the model for the math QA task due to time and computing power constraints, thus excluding it from the ablation results.
>
> For the story generation task, we did not do an ablations study on it because of our negligence. We only focus on conducting experiments on story generation for the main correlation results. It's a great suggestion to include them in the ablation study for better comparison.
>
> Therefore, in the current version, in response to your suggestion, we have included all the results of mathQA and story generation in the ablation studies. Results show that the combination of real-world data and synthetic data, along with mixed training on all tasks, has been found to enhance the performance of TIGERScore in most cases compared to individual training. You can refer to more details in Section 4.5 of our revised paper version.
>
> We hope our response answers your concern well!

---

> ### Author Response · Authors · 2023-11-22
> **Response to Question 7 about training strategies**
>
> > Is there any curriculum used for training on the different tasks?
>
> No, we simply mix the data from all the tasks and perform the default random sampling during the training process.
>
> One idea that might be interesting, but we did not try due to the limited resources, is first finetuning on the first 5 well-defined tasks, i.e. summarization and translation, etc, and then do the instruction-finetuning on the instruction-following dataset. We leave this for further research.

---

> ### Author Response · Authors · 2023-11-22
> **Response to Question 8 about code availability**
>
> > Would the corresponding code be made available which is necessary for reproducibility.
>
> Thank you for your interest in the TIGERScore code repository. We are committed to the principles of open-source community conduct and have made all the code available on GitHub.
>
> You can find further details about our project on the [website](https://tiger-ai-lab.github.io/TIGERScore/) listed in our paper. Our efforts have been focused on enhancing the user-friendliness and accessibility of our code repository. Additionally, we are in the process of integrating TIGERScore into the Hugging Face evaluation metrics to better serve the research community.

---

> ### Author Response · Authors · 2023-11-22
> **Response to the paper writing suggestions**
>
> > Suggestions/Comments: Fig 1 caption: shows te -> shows the Section 5.2: Research on this fields -> in these fields Please take care of the quotes in Section 2, Table 3 caption Please specify and distinguish between LLaMA and LLaMA2 in the abstract and the main paper.
>
> Thank you for your writing suggestions. We have corrected the typo in Figure 1, the grammatical error in Section 5.2, and the issue with quotes in Section 2. Additionally, we have updated all mentions of 'Llama' in the paper to 'Llama-2', which is the specific version TIGERScore is fine-tuned on.

---

### Official Review · Reviewer_WhZB · 2023-11-01

**Soundness:** 3 good
**Presentation:** 2 fair
**Contribution:** 3 good
**Rating:** 8
**Confidence:** 4

**Summary:**

The authors proposed TIGERscore, a reference-less evaluation metrics for text generation tasks that improves the interpretability of the evaluation metrics by generating itemized, structured errors with information on location, explanation/correction suggestion, and a penalty score for each error. TIGERScore is built on LLAMA foundation models finetuned with MetricInstruct dataset, in which each sample consists of instruction, context, and output from some system as Input, and a list of structured errors with all aforementioned information as Target. The MetricInstruct dataset is the key to the good performance of TIGERSCORE, and is derived from 23 distinctive public NLG datasets covering six tasks, and the target structured errors are generated by querying GPT-3.5-Turbo and GPT-4 models with heuristic filtering of the candidate errors. On five out of seven test sets, TIGERScore has outperformed a wide range of reference-based and reference-less metrics in terms of human correlation, and additional human evaluation has confirmed the majority of the provided structured errors by TIGERScore to be reasonable and correction suggestions to be helpful.

**Strengths:**

Originality:

Similar works have been done on related topics of creating an evaluation metrics with a foundation model backbone that provides more than just a final score, but TIGERScore's way of constructing the additional evaluation output (the structured errors: location, explanation, suggestion, and penalty) is original in the sense of the level of granularity at which people have attempted in the design of a highly interpretable evaluation metrics.

Quality:

The overall analysis is solid, the choice of using correlational analysis with human evaluation score is appropriate and to the point (to aim for an evaluation metrics that emulate human judgement), and an extensive selection of both reference-based and reference-less metrics have been provided as baselines, ensuring the resulting performance of the proposed metrics to be convincing as well as up-to-date.


Clarity:

The paper is generally very easy to follow with comprehensive examples and illustrations provided in both the main text and the appendix. There are places with typos and possibly incorrect sentences, for which the reviewer will provide a list in **Questions** below.

Significance:

The importance of having a robust, interpretable, and relevant evaluation metrics for text generation task is beyond doubt in current NLP research (see for example: https://arxiv.org/abs/2202.06935), and this paper has offered not only a competitive reference-less evaluation metrics, but also a valuable dataset (MetricInstruct) for furthering research on more robust and versatile evaluation metrics in the community. The significance is clearly high.

**Weaknesses:**

The major weakness of the paper is the lack of discussion on the distribution of quality in the provided dataset. The authors explained clearly their efforts in ensuring coverage of error types, but there is no clear discussion on how the quality of the system outputs studied in the test sets is controlled. For example, it is unclear how TIGERScore would perform on notes with no errors. It is also unclear if an output with a higher TIGERScore would actually reflect higher human preference in an A/B testing or a ranking scenario. Also, the authors didn't provide an exhaustive list of error types ($E_i$) used in their model training, therefore making the generalizability of the score to possible unseen error types questionable.

Another weakness is the clarity in the writing, the overall language is very easy to follow, but there are some problems with typos, non-sensical words, incorrect sentences, and orphaned table/figure. Please see **Questions** below for a non-exhaustive list.

**Questions:**

General Questions:
1. How many system outputs are actually correct (no errors)? How well does TIGERScore perform on those outputs? Would it hallucinate errors frequently? Have you tried running TIGERScore on reference outputs?
2. How are the prompt templates in Appendix 2 created? There are minor variations from task to task, is this intentional (e.g., the way minor and major mistakes are explained in the prompts changed from template to template)?
3. Why explaining major and minor mistakes in the prompts, when you can ask the model to infer from the aspect list? Is this for providing two criteria (major/minor label, and penalty score) to filter unreasonable candidate errors?
4. Some of the templates (e.g., Table 11) are missing prefixes like "References:", "Task instructions:", is this intentional or a typo?
5. Why is $aspect_list not used in translation and data2text prompt templates?
6. Are the definitions of evaluation aspects ($aspect_list) for each task created based on consensus or by the authors?
7. Why is a scale from 5 to 0.5 chosen? Any special reason?

Editing Suggestions:
1. Figure 2 and Table 5 are never referred to in the main text, please fix this.
2. Section 1, *...Our analysis shows that MetricInstruct is attributed to three key aspects in MetricInstruct: ...*; Probably the first *MetricInstruct* should be sth like "success of TIGERScore"?
3. Section 2.3, although **Training Setup** in Section 4.2 has explained what the model behind TIGERScore is, it is arguably better to include an explanation of the backbone model and training procedure here (maybe in section 2.4). Otherwise it reads like the method section doesn't actually explain what the method itself is.
4. Section 3.2, *...carefully designed prompting templates...*; you can make reference to tables in Appendix A.2 here.
5. Section 3.3, *...we employ Levenshtein Distance to check whether the illusory location in analyses and exclude evaluations with a low token set ratio...*; I find this sentence very hard to understand, how is *token set ratio* defined for Levenshtein distance (the former disregards order while the latter respects it explicitly)? Do you mean low LD score? So the Levenshtein distance used is calculated on characters, or tokens, or words? If tokens, what tokens, tokens generated by the tokenizer of the model? Also *check whether the illusory location in analyses* is an incomplete sentence, perchance you mean *check whether there is illusory location in the analysis*?
6. Section 4.1, *...System outputs to be evaluated of each test dataset either from official releases...*; again, incomplete sentence (there is no verb here), please rewrite
7. Section 4.3, ***...Nota bene*** *that TIGERSCORE significantly surpasses traditional metrics...*; typo?
8. Section 4.3, *...This suggests that TIGERSCORE can effectively evaluate text generation models even when the tasks have not been seen during training...*; Please consider rewriting this sentence or elaborate with more analysis, TIGERScore performs very weak on StoryGen (lower end among reference-based metrics and bottom one among reference-less metrics), so it is a stretch to claim it performs even reasonably well on unseen tasks.
9. Table 5, second column, *...Explaination Error?...*; typo, please fix it.
10. Section 4.3, *...In 70.6% of the error analyses, no errors were found to be missing...*; I assume 70.6% was calculated based on the number of selections with a score 3 or 4 by human; this is not rigorous given the example answer options in Appendix A.3: 3 still corresponds to explanations with missing errors (arguably less critical, but still). This is a common problem with some of the percentages referred to later in the same section, please state clearly how those percentages were calculated, and explain the reason when scores less than the highest value for each category are used (e.g., it seems 3 to 5 are used for calculating 70.8% for Overall Rating).

---

> ### Author Response · Authors · 2023-11-22
> **Response to Reviewer WhZB**
>
> Dear Reviewer `WhZB`:
>
> Thank you for your insightful review. We are delighted to hear that you recognize TIGERScore as a competitive reference-less evaluation metric of high significance.
>
> According to the feedback of all the reviewers, we have added the corresponding experiments in the [**revised paper**](https://openreview.net/pdf?id=SIojR1ruNQ) with supplementary experiments in the appendix. We have summarized the major updates in  `General Response` (shown in a separate comment on this page).
>
> ***We would like to address your concerns in detail below.***

---

> > ### Comment · Reviewer_WhZB · 2023-11-22
> > **Thank you very much for the rebuttal responses!**
> >
> > I have read through Appendix 1, 2, 5, and 7, and thank you very much for addressing my questions with direct quantitative results and much appreciated for the additional efforts in organizing those results nicely in the appendix.
> >
> > It does shown from Table 21 that TIGERScore may be intrinsically biased towards penalization even in case of "perfect" output, this is an aspect of TIGERScore that readers should be made aware of, but this doesn't necessarily diminish its value or potential applications in realistic scenarios, when errors are expected from machine-generated output across various tasks.
> >
> > With the edited and added contents, I think this paper has improved in robustness and I believe it should be accepted.

---

> ### Author Response · Authors · 2023-11-22
> **Response to Weakness 1 about dataset distribution**
>
> > Weakness 1: lack of discussion on the distribution of quality in the provided dataset. For example, it is unclear how TIGERScore would perform on notes with no errors or a ranking scenario.
>
> Thank you for your valuable suggestion regarding the analysis of our dataset distribution. In response, we have included a comprehensive analysis in Appendix A.5. This addition aims to enhance our understanding of MetricInstruct for our users.
>
> Specifically, we have analyzed the context lengths of training instances to determine the most appropriate model context length. Furthermore, we have investigated the frequency of errors in each instance across all tasks. Our findings indicate a natural distribution of errors. It’s crucial that a number of instances with no errors exist in the dataset because it helps the model learn what is correct and reduces fabricated errors from the model.

---

> ### Author Response · Authors · 2023-11-22
> **Response to Question 1 about hallucination analysis**
>
> > Question 1: How many system outputs are correct (no errors)? How well does TIGERScore perform on those outputs? Would it hallucinate errors frequently? Have you tried running TIGERScore on reference outputs?
>
> Thank you for your insightful question. We'll address your queries sequentially.
>
> To begin with, referencing our earlier response to weakness 1, you'll find a detailed analysis in Appendix A.5. **The data indicates that approximately 20% of the system outputs in the training dataset are entirely error-free.**
>
> Furthermore, to assess the effectiveness of TIGERScore on correct outputs and its potential for generating hallucinated errors, we have constructed a collection of reference outputs to create new test datasets for each task, as you recommended. Results are summarized below (or see Appendix A.7 Table 21).
>
> Here for each task, the $0$ column refers to the percentage that TIGERScore reduces 0 scores for the gold references. The $>-2$ column refers to the percentage that TIGERScore reduces less to 2 score on the gold references.
>
>
> | Tasks$\rightarrow$                                 | Summarization |       | Translation |       | Data2Text |       | Long-form QA |       | MathQA |       | Inst-Fol |       | Story-Gen |       | Average |           |
> | -------------------------------------------------- | :-----------: | :---: | :---------: | :---: | :-------: | :---: | :----------: | :---: | :----: | :---: | :------: | :---: | :-------: | :---: | :-----: | :-------: |
> | Models$\downarrow$Proportion of Score$\rightarrow$ |       0       |  >-2  |      0      |  >-2  |     0     |  >-2  |      0       |  >-2  |   0    |  >-2  |    0     |  >-2  |     0     |  >-2  |    0    |    >-2    |
> | TigerScore-7B-v1.2                                 |     16.00     | 68.00 |    3.57     | 72.48 |   45.51   | 78.65 |    84.75     | 84.75 | 34.36  | 34.36 |  73.98   | 92.48 |   34.00   | 34.00 |  41.74  | **66.39** |
> | TigerScore-13B-v1.2                                |     48.00     | 97.00 |    21.01    | 83.63 |   23.03   | 94.94 |    94.50     | 94.50 | 25.28  | 25.28 |  86.38   | 96.14 |   46.00   | 49.00 |  49.17  | **77.21** |
>
> On average, TIGERScore (13b) does not produce any errors in the gold reference sentence about **77.21%** of the time. In these instances, it either makes no errors or only minor ones, resulting in a score reduction of less than 2.0 (minor error). **Therefore, while TIGERScore may occasionally hallucinate errors, its overall rate of hallucination is deemed acceptable.**
>
> Besides, for each task, we ran TIGERScore on 20 samples with errors in the system output. We then used GPT-4 to determine if these samples contained hallucinations or factual inaccuracies (see Appendix A.7).
>
> | Tasks    | Summarization | Translation | Data2Text | Long-form QA | MathQA | Inst-Fol | Story-Gen | Total |
> | -------- | :-----------: | :---------: | :-------: | :----------: | :----: | :------: | :-------: | ----- |
> | Accuracy |     95.00     |    95.00    |   90.00   |    85.00     | 90.00  |  75.00   |   95.00   | 89.28 |
>
> According to the results above, approximately 89.28% of TIGERScore's error analyses were free from hallucinations or factual errors. We acknowledge the limitations of our study, including the small sample size and the reliance on GPT-4 rather than human evaluators. **Nonetheless, our findings are significant, demonstrating that TIGERScore is effective at avoiding hallucinations in generated content.**
>
> We hope our response answers your concern well!

---

> ### Author Response · Authors · 2023-11-22
> **Response to Q2-Q7 about the prompting templates design**
>
> Thank you for your inquiries regarding our prompting template design. Given that questions Q2 through Q7 all relate to this topic, we have opted to collectively address them in Appendices A.1 and A.2, where we discuss the various prompting strategies employed in the template creation process. Below, we provide a summarized response to each question.
>
> > Question 2: How are the prompt templates in Appendix 2 created? There are minor variations from task to task, is this intentional (e.g., the way minor and major mistakes are explained in the prompts changed from template to template)?
>
> The templates discussed in this paper were developed from our practical experience and intuition, as detailed in Appendix A.2. They all stem from a single base template, with minor variations introduced to incorporate task-specific terminology, aiming to influence GPT-4's responses in a certain way.
>
> Furthermore, to assess the effectiveness of these templates, we conducted preliminary experiments focusing on their correlation. During the development process, each template underwent several iterations by a designated author, leading to the final versions presented in this paper.
>
> >Question 3: Why explain major and minor mistakes in the prompts, when you can ask the model to infer from the aspect list? Is this for providing two criteria (major/minor label, and penalty score) to filter unreasonable candidate errors?
>
> This explanation is inspired by the MQM translation human rating system and InstructScore prompting template. Our intuition is that this classification helped GPT-4 in assigning more consistent scores to each error, countering its instability with numerical judgments.
>
> > Question 4: Some of the templates (e.g., Table 11) are missing prefixes like "References:", and "Task instructions:", is this intentional or a typo?
>
> No, As stated in our response to question 2 and Appendix A.2, these templates are the results of a few iterations by our authors, together with some intuitions. For example, in tasks like mathQA and instruction-following, where the context is self-explanatory, we omitted specific keywords like “Task Instruction” and “reference”.
>
> > Question 5:. Why is $aspect_list not used in translation and data2text prompt templates?
>
> As stated in Appendix A.2, the incorporation of ${aspect_list} is used to address the issue where GPT-4 might miss crucial evaluation aspects and overlook some errors that we think are important. Exceptionally, in data2text tasks, we found that asking GPT-4 to directly evaluate errors was sufficient, and thus, we did not include our predefined aspects.
>
> > Question 6:. Are the definitions of evaluation aspects ($aspect_list) for each task created based on consensus or by the authors?
>
> As stated in Appendix A.1, these aspects are created with the assistance of GPT-4 and human revision. They are designed to be mutually exclusive and collectively exhaustive for a single task. The steps to create these aspects are:
>
> - Step 1: We prompt GPT-4 to output 20 candidate aspects for each task.
> - Step 2: We ask GPT-4 to summarize these aspects into 3 to 5 general aspects for this task.
> - Step 3: We ask GPT-4 to generate the detailed definition and 5 specific error types under each aspect.
>
> Then after final human revision by our authors, these aspects are created.
>
> > Question 7:. Why is a scale from 5 to 0.5 chosen? Any special reason?
>
> In Appendix A.2, we initially employed an integer scale ranging from 1 to 5 for error penalty scores. This scale proved effective for translation tasks. However, for other tasks such as summarization, data-to-text generation, and instruction-following, it was less effective. Our experiments showed that a more nuanced scoring scale, ranging from 0.5 to 5, resulted in a better correlation across all tasks.

---

> ### Author Response · Authors · 2023-11-22
> **Response to the editing suggestions**
>
> Thanks for your valuable suggestions on the writing!
>
> - In response to your suggestion 1, we have referred to Figure 2 and Table 5 in the corresponding text sections that discuss their results.
>
> - We have addressed suggestions 2, 6, 7, and 9 by correcting the typographical errors.
>
> - Following your suggestion 3, we have moved the original Section 4.2 to Section 2.4, adding more details about TIGERScore's implementation in Section 2 as you requested.
>
> - As per your suggestion 4, we have included references to the tables immediately following this sentence.
>
> - For your suggestion 5, We have revised this sentence and excluded the existence of LD distance to make it more clear. The revised one is:
>
> > In advance, to mitigate the impact of the hallucination in error analysis, we split the error location and the source by spaces. Then we remove error positions that contain too many words that do not appear in the input source to avoid the illusion of error locations.”
>
> - In line with suggestion 8, we have incorporated the results of TIGERScore V1.2. This update demonstrates a significant enhancement in StoryGen's performance, underscoring our improved generalization capabilities. Therefore, we believe the statements still hold well.
>
> - For suggestion 10, we have added simple phrases after each number indicating how we calculated them.

---

### Official Review · Reviewer_2ZJr · 2023-11-04

**Soundness:** 3 good
**Presentation:** 3 good
**Contribution:** 2 fair
**Rating:** 5
**Confidence:** 4

**Summary:**

This paper introduces TIGERScore, a novel evaluation methodology designed for natural language generation (NLG) tasks. The primary concept involves creating a new training dataset containing human preference scores sourced from existing NLG datasets. The authors augment this dataset by generating synthetic scores using GPT-4. Additionally, they leverage GPT-4 to produce error explanations that delineate the facets of errors. Training TIGERScore involves utilizing LLaMA-2. As a reference-free evaluation method, TIGERScore solely requires instruction, context, and the model output for evaluation. Experimental results demonstrate that TIGERScore exhibits a high Spearman’s correlation with human preference scores in held-in evaluation datasets and some held-out evaluation datasets. Through human expert ratings, TIGERScore offers explanations for the assigned scores. An ablation study highlights the impact of the synthetic dataset and the synergy achieved by combining multiple tasks in contrast to training with only one task.

**Strengths:**

The primary strength of this paper lies in proposing a robust and adaptable NLG evaluation methodology. It utilizes several NLG datasets as a training dataset and operates as a reference-free methodology. This characteristic enables other AI and NLP researchers to extend this concept for evaluating their individual NLG models.

**Weaknesses:**

This paper raises two concerns: overclaim and human evaluation. TIGERScore exhibits limitations, particularly evident in held-out tasks (story generation), raising doubts about the claim of being a universal metric for all text generation tasks in the title. The title prompts questions about its efficacy in especially evaluating open-domain conversation tasks, given the considerable diversity in model outputs.

Regarding human evaluation, there's ambiguity about the identity of the 'human experts' and the reliability of their responses. And the overall rating of TIGERScore is noted as 70.8%, but it's unclear whether this score is deemed acceptable or not. It would be beneficial to compare TIGERScore with other existing explainable metrics in Section 5.2, demonstrating the differences from the baselines.

**Questions:**

- Have other correlation metrics, like Pearson and Kendall rank correlation coefficients, been considered? These metrics might reveal different aspects of the correlation between model outputs and human preferences.
- Could you clarify who the 'human experts' are for the human evaluation?
- It would enhance clarity to cite the paper when introducing terms for the first time, such as ASQA and FeTaQA terms in section 3.1.
- Errata: Replace "te" with "the" in the Figure 1 caption.

---

> ### Author Response · Authors · 2023-11-22
> **Response to Reviewer 2ZJr**
>
> Dear Reviewer `2ZJr` :
>
> Thank you for your insightful review. We are glad to hear that you appreciate TIGERScore as a robust and adaptable NLG evaluation methodology.
>
> According to the feedback of all the reviewers, we have added the corresponding experiments in the [**revised paper**](https://openreview.net/pdf?id=SIojR1ruNQ) with supplementary experiments in the appendix. We have summarized the major updates in  `General Response` (shown in a separate comment on this page).
>
> ***We would like to address your concerns in detail below.***

---

> > ### Author Response · Authors · 2023-11-22
> > **Response to Q3 and Q4 on the paper writing**
> >
> > >Q3: It would enhance clarity to cite the paper when introducing terms for the first time, such as ASQA and FeTaQA terms in section 3.1.
> >
> > >Q4: Errata: Replace "te" with "the" in the Figure 1 caption.
> >
> > Thanks for your paper writing suggestions! We have added the corresponding citations for both ASQA and FeTaQA in section 3.1. And fix the typo "te" in the caption of Figure 1

---

> ### Author Response · Authors · 2023-11-22
> **Response to Weakness 1 about correlation performance**
>
> > TIGERScore exhibits limitations, particularly evident in held-out tasks (story generation), raising doubts about the claim of being a universal metric for all text generation tasks in the title. The title prompts questions about its efficacy in especially evaluating open-domain conversation tasks, given the considerable diversity in model outputs.
>
> Thanks for pointing out the concern of TIGERScore’s limitations in held-out tasks and we totally understand your concern. We noticed these weaknesses after the V1 development and have always been investigating the reason behind them.
>
> It’s shown that the TIGERScore-V1’s limitations on generalization ability are caused by the lack of diversity of tasks in the datasets. Though the MetricInstruct V1 has included various text generation tasks and broad error types in training instances, they are still limited to a specific one, except for the instruction-following task.
>
> In TIGERScore V1.2, **we value the importance of instruction-following tasks, which we believe is important in increasing the generation ability of TIGERScore**. Therefore, we collect 10k more instruction-following data from alpaca-52k and apply the same data preparation pipeline querying GPT-4 to get high-quality error analysis responses. We train TIGERScore-V1.2 on them and the results show that the generalization ability has been increased greatly.
>
> We now have trained a new version of TIGERScore-V1.2, which gets great performance enhanced in generalization ability, shown by the increased correlation score in the story generation task. TIGERScore V1.2’s results have been updated in Table 4. A summarized difference between V1.
>
> |            Tasks$\rightarrow$             | Summarization | Translation | Data2Text  | Long-form QA |  MathQA   | Inst-Fol  | Story-Gen |  Average  |
> | :---------------------------------------: | :-----------: | :---------: | :--------: | :----------: | :-------: | :-------: | :-------: | :-------: |
> | Metrics$\downarrow$ Datasets$\rightarrow$ |   SummaEval   | WMT22-zh-en | WebNLG2020 |    ASQA+     |   gsm8k   |   LIMA+   |    ROC    |           |
> |            TigerScore-7B-V1.0             |   **45.52**   |    34.52    |   50.35    |    42.45     | **33.44** |   26.97   |   29.97   |   37.60   |
> |            TigerScore-13B-V1.0            |     45.28     |  **41.70**  |   49.02    |  **45.91**   |   30.68   |   36.92   |   21.83   |   38.76   |
> |            TigerScore-7B-V1.2             |     43.95     |    37.70    |   49.13    |    46.10     |   21.77   |   38.26   |   39.90   |   39.54   |
> |            TigerScore-13B-V1.2            |     44.21     |    41.54    | **52.87**  |    44.76     |   24.41   | **47.52** | **47.66** | **43.28** |
>
> From the table, we can see **clear improvements in data2text, instruction-following, and the held-out story-gen task**. And still **maintain similar performance with V1 on summarization, translation, and long-form QA tasks**.

---

> ### Author Response · Authors · 2023-11-22
> **Response to Weakness 2 and Question 2 about Human eval**
>
> > Weakness 2: Regarding human evaluation, there's ambiguity about the identity of the 'human experts' and the reliability of their responses. The overall rating of TIGERScore is noted as 70.8%, but it's unclear whether this score is deemed acceptable or not. It would be beneficial to compare TIGERScore with other existing explainable metrics in Section 5.2, demonstrating the differences from the baselines.
>
> > Question 2: Could you clarify who the 'human experts' are for the human evaluation?
>
>
> Thank you for your constructive suggestion!
>
> To clarify, "human experts" refers to evaluators chosen from Prolific, a renowned crowdsourcing platform. These individuals must pass certain tests to be approved as official annotators on Prolific. We maintain high evaluation standards by periodically sampling and reviewing these annotations. If any annotation is found to be of poor quality, it is removed from our system along with all other work by the responsible annotator. This practice is key to upholding the integrity of our evaluations.
>
> Regarding the TIGERScore, as mentioned in Section 1, it stands out as one of the few metrics that is both explainable and applicable across various tasks. Unlike metrics like UniEval and GPTScore, which are considered explainable due to their multi-aspect scoring capabilities but lack text explanation capabilities, TIGERScore offers contextual text explanations. While PandaLM can generate text explanations, they are limited to simplistic overall assessment sentences, providing minimal information. Moreover, InstructScore is confined to translation tasks, making it non-comparable across different tasks. **Due to these distinct features, we did not conduct a direct comparison of explanation quality with baseline metrics**.
>
> Lastly, the 70.8% overall positive rating pertains specifically to the feedback that  "TIGERScore’s explanations provide helpful revision suggestions." This suggests that TIGERScore's explanations are perceived as beneficial around 70.8% of the time. While determining if a 70.08% rating is acceptable can be subjective and lacks a comparative context, we believe this percentage does affirm the absolute helpfulness of TIGERScore.

---

> ### Author Response · Authors · 2023-11-22
> **Response to Question 1 about adding Pearson and Kendall**
>
> > Q1: Have other correlation metrics, like Pearson and Kendall rank correlation coefficients, been considered? These metrics might reveal different aspects of the correlation between model outputs and human preferences.
>
> Thank you for your valuable suggestions. We did record the results of Pearson and Kendall correlation results in all of our experiments. To provide a comprehensive view, we have included the correlation results of all three metrics (Pearson, Kendall, and Spearman) in Appendix A.7. Our findings show that the **results from Pearson and Kendall analyses are consistent with those from Spearman reported before**, affirming that TIGERScore correlates effectively with human ratings across various correlation aspects.

---

### Author Response · Authors · 2023-11-22
**General Response**

Dear reviewers,

We are grateful for all your insightful reviews and are glad that you have recognized TIGERScore's strengths. We briefly summarize them as follows:

- **Robust reference-free and explainable metric**: "a competitive reference-less evaluation metric", "robust and adaptable NLG evaluation methodology.", "transparent and trustworthy".
- **Valuable Dataset + Novel collection Strategies**:  "a valuable dataset (MetricInstruct)", "interesting for the community", "furthering research on more robust and versatile evaluation metrics"
- **Clear writing**: "easy to follow", "comprehensive examples and illustrations",

### Please check the [revised paper version](https://openreview.net/pdf?id=SIojR1ruNQ) for detailed updates for rebuttal

****

We thank all the reviewers for their constructive feedback and valuable questions. To respond to the concerns raised in the comments, we have submitted a revised version of our paper with improved writing and supplementary experiments in the appendix. New contents section titles or contents are marked in blue color for better clarity. We summarize the major updates in the following:

### 1. Enhanced version of TIGERScore V1.2

- To resolve the concern about TIGERScore's generalization ability (especially in held-out task story generation) from reviewer `2ZJr`, `Az3X`, and `A6PK`, we deliver TIGERScore V1.2 with enhanced correlation performance on all tasks, approaching the results of GPT-4-0-shot.
- We discuss major updates of TIGERScore V1.2 compared to V1 in `Appendix A.7`, where we mainly collect more data and adopt better data filtering strategies.

### 2.  Additional experiment results

- As suggested by reviewer `2ZJr`, we add correlation results on Pearson and Kendall as complementary results to the Spearman metric in `Appendix A.7`.
- As suggested by reviewers `WhZB` and `Az3X`,  we add experiments to investigate the hallucinations of TIGERScore to make it more trustworthy in `Appendix A.7`.
- As suggested by reviewer `Az3X`, we add more powerful baselines including Llama-2-Chat-0-shot and GPT-4-0-shot in `Table 4` and `Appendix A.7`.

### 3. Supplementary Analysis and Discussion Sections

- As suggested by reviewer `WhZB`, we add a detailed analysis of the dataset distribution to help users know better about MetricInstruct in `Appendix A.5`.
- As suggested by reviewers `WhZB` and `A6PK`, we add a comprehensive discussion of the prompting template design strategies and intuitions in `Appendix A.1`  and `Appendix A.2`
- In response to the suggestions from reviewers `Az3X` and `A6PK`, we have included a new section discussing the limitations of TIGERScore in `Appendix A.6`. This section focuses on scenarios where TIGERScore might not be effective and its computational efficiency, providing clearer guidance on the most appropriate situations for its use.

---

### Thank you!

We sincerely thank the reviewers for their constructive suggestions and questions to enhance our paper. We think that our rebuttal has sufficiently addressed the concerns raised by the reviewers. Please reply if you have any further questions, and we will be more than happy to continue the discussion.

---

### Meta-Review · Area_Chair_BNsK · 2023-12-10

**Metareview:**

This paper introduces TIGERScore based on LLaMA 2 for NLG evaluation. For training this metric using natural language instructions, the MetricInstruct dataset is introduced, containing 48k examples and covering six text generation tasks (Summarization, Translation, Data-to-text, LongFormQA, MathQA, and Instruction Following) across 23 datasets.

There are widespread scores among the reviewers (two rejections and two acceptances). The reviewers admit the novelty and writing of this paper. However, there are also too many concerns about the details and clarity. The authors have tried their best to address the reviewers' concerns and one reviewer is convinced. However, the other reviewers maintained their score.

In all, the paper needs significant improvement and cannot be accepted in the current version.

**Justification For Why Not Higher Score:**

the paper needs significant improvement and cannot be accepted in the current version.

**Justification For Why Not Lower Score:**

n/a

---

### Decision · Program_Chairs · 2024-01-16

Reject